# Three-component contour dynamics model to simulate and analyze amoeboid cell motility in two dimensions

Daniel Schindler[1,3], Ted Moldenhawer[2,3], Carsten Beta[2,3], Wilhelm Huisinga[1,3], Matthias Holschneider[1,3] *

1 Institute of Mathematics, University of Potsdam, Potsdam, Germany, 2 Institute of Physics and Astronomy, University of Potsdam, Potsdam, Germany, 3 CRC 1294 Data Assimilation, University of Potsdam, Potsdam, Germany

* hols@uni-potsdam.de

**Data Availability Statement:** The source code and the minimal data set (contour data and microscopy data) are available within our software package AmoePy in two seperate folders. URL: https://

## Abstract

Amoeboid cell motility is relevant in a wide variety of biomedical processes such as wound healing, cancer metastasis, and embryonic morphogenesis. It is characterized by pronounced changes of the cell shape associated with expansions and retractions of the cell membrane, which result in a crawling kind of locomotion. Despite existing computational models of amoeboid motion, the inference of expansion and retraction components of individual cells, the corresponding classification of cells, and the *a priori* specification of the parameter regime to achieve a specific motility behavior remain challenging open problems. We propose a novel model of the spatio-temporal evolution of two-dimensional cell contours comprising three biophysiologically motivated components: a stochastic term accounting for membrane protrusions and two deterministic terms accounting for membrane retractions by regularizing the shape and area of the contour. Mathematically, these correspond to the intensity of a self-exciting Poisson point process, the area-preserving curve-shortening flow, and an area adjustment flow. The model is used to generate contour data for a variety of qualitatively different, e.g., polarized and non-polarized, cell tracks that visually resemble experimental data very closely. In application to experimental cell tracks, we inferred the protrusion component and examined its correlation to common biomarkers: the F-actin density close to the membrane and its local motion. Due to the low model complexity, parameter estimation is fast, straightforward, and offers a simple way to classify contour dynamics based on two locomotion types: the amoeboid and a so-called fan-shaped type. For both types, we use cell tracks segmented from fluorescence imaging data of the model organism *Dictyostelium discoideum*. An implementation of the model is provided within the open-source software package AmoePy, a Python-based toolbox for analyzing and simulating amoeboid cell motility.

## Introduction

Amoeboid movement belongs to the most widespread kinds of eukaryotic cell motility [1, 2]. It plays a key role in many biophysical and physiological processes such as wound healing,

zenodo.org/record/3982371 DOI: 10.5281/zenodo.3982371 Source code folder: B02-AmoePy.zip minimal data set folder: b02-data.zip.

**Funding:** The research of DS and TM has been partially funded by the Deutsche Forschungsgemeinschaft (DFG)- Project-ID 318763901 - SFB1294. The funders had no role in study design, data collection and analysis, decision to publish, or preparation of the manuscript. There was no additional external funding received for this study.

**Competing interests:** The authors have declared that no competing interests exist.

cancer metastasis, and immune system responses and is characterized by dynamic changes of the cell shape [3–5]. In this context, the cell is moving forward by creating protrusions, so-called pseudopodia. The location of these protrusions and the frequency of their formation mainly define the overall trajectory of the cell [6]. In addition to these explorative and faster membrane protrusions, the cell retracts its rear (so-called uropod) in a slower and steadier way. By this coordinated interplay of protrusions and retractions, the cell can move persistently and efficiently in a crawling-like fashion. Finally, the cell track is affected by the creation and rupture of adhesion contacts to the substrate [7].

On an intracellular level, amoeboid migration is initialized by changes of the actin cytoskeleton, a microfilament meshwork consisting of many multi-functional proteins. More precisely, the cytoskeleton grows by polymerization and shrinks by depolymerization of actin filaments. This rearrangement of the actin network is controlled by additional cytoskeletal proteins initializing capping, severing, and nucleation of actin filaments creating different meshwork formations such as bundling, cross-linking, and branching [8]. The underlying mechanics are regulated by biochemical signaling pathways initializing different actions such as proliferation by cell growth and cell division, cell aggregation, and coordinated cell death [9, 10]. Furthermore, signaling pathways are linked to membrane receptors, enabling the cell to move persistently and directly towards a chemical source (chemotaxis) by sensing gradients and extracellular cues [11].

The formation of pseudopodia is the key aspect of amoeboid motility. The size and shape of these pseudopodia can vary significantly for different cell types [12]. Some organisms such as *Dictyostelium discoideum* even possess different modes of locomotion and can switch back and forth between them [13]. Therefore, pseudopodia are subdivided based on their nature into different types: lobopodia, lamellipodia, filopodia, reticulopodia, axopodia, invadopodia, and others [14–16]. Pseudopodia can be also classified depending on the exact location of their appearance: Y-shaped (split) pseudopodia at the front, and *de novo* pseudopodia at the side or rear of the cell changing the direction of its movement [17]. The simulated contour dynamics within this work possess protrusions mostly resembling lobopodia, i.e., cylindrical and finger-shaped protrusions, and lamellipodia, i.e., protrusions with a flat and broader structure [12]. However, the model is not limited to these two protrusion types due to the versatile parameter regime to control the size, speed, duration, and number of protrusions.

A wide variety of amoeboid motility models have been proposed tackling different aspects of amoeboid migration such as membrane protrusion and retractions, trajectories of the cell's centroid, its polarization, and the influence of chemoattractant cues. Many proposed models are based on the concentration of interior biochemical compounds of the cell [18, 19]. In these reaction-diffusion models, motility patterns often result directly from the interplay of different processes controlling local excitation and global inhibition of the intracellular signaling and cytoskeletal dynamics [20–23]. Reaction-diffusion models have been used to describe the self-organized polarization of the cell in the presence of chemoattractants [24–28], and the occurrence of intracellular waves and oscillations [29–33]. A large number of these approaches are based on phase-field models which are used to describe the transition between different phases such as liquid and solid states or the interior and exterior of the cell. In the latter case, one modeling approach is to define the interior and exterior of the cell as binary states with a smooth transition function. The cell membrane is then defined where the transition function reaches the exact midpoint of both states [34]. Phase-field models are often used to describe *D. discoideum* [13, 26, 34–36], but also for other cell types [37–41].

Other approaches are mechanical models in which different forces affect the cell's motility from within and from the outside. Mechanical models differ in complexity and dimensionality to target different problems, e.g., the formation of fibroblasts (1D) [42], the influence of

contraction and adhesion sites to the substratum on amoeboid cell motility (2D) [43], and the evolution of the cell surface obtained from triangulation under chemotaxis (3D) [44]. Furthermore, different physical methods and assumptions are used for the underlying model equations such as active gel physics where the gel consists of polymer filaments permeated by a solvent [45], hydrodynamics to model the internal cytoplasmic fluid [46], and the modeling of the extra-cellular matrix [47, 48]. Most importantly, mechanical models can be easily adjusted for unusual types of cell migration such as amoeboid swimmers [49, 50]. In [51], a mechano-chemical model is proposed for which the underlying parameters are calibrated by using Bayesian optimization. Additionally, extensions of the cellular Potts model have been successfully used to reproduce amoeboid and keratocyte-like movements [52, 53].

Finally, level set methods are used to simulate cell tracks, e.g., as part of a mechanical model of the cell cortex combined with an excitable network acting as an activator/inhibitor system [19]. The excitable network is triggered by random fluctuations and can be enhanced with additional gradient stimuli and polarization modules [54, 55]. In [56], the stochastic extension of pseudopods during chemotaxis is modeled. Furthermore, stochastic differential equations have been used to model the trajectory of the cell's centroid, e.g., by a (generalized) Langevin equation [57–61]. In [62], the centroid trajectory was modeled by a Markov chain approach on a discrete domain obtained from hexagonal tiling. Geometric equations of evolving curves are commonly used to describe cell migration [23, 44, 63, 64]. A novel contour evolution method based on the curve-shortening flow (CSF) or, alternatively, with an optional additive function which is then called forced CSF, is presented in [65] and then applied to cell migration in [66]. Usually, the cell contour is parametrized by the contour arc length [67–69] or polar coordinates relative to the center of mass [70]. A comparison of the different model approaches mentioned above can be found in [71–74].

In contrast to the above approaches, our model is designed to infer key characteristics of individual cell tracks, i.e., the intensity of protrusions and retractions during amoeboid cell motility. The model is therefore intended to be simple and intuitively comprehensible to ensure a fast and straightforward estimation of underlying model parameters and to be capable of producing a specific motility behavior for a given input. Especially the second property is sometimes hard to achieve with mechanical models, which tend to have a higher complexity with a large number of model parameters and entangled subprocesses. This makes it difficult to draw a direct link between model parameters and a desired motility outcome [48]. By inferring the above characteristics, our model can also be used to identify differences between cells, to classify them, and as a general comparison tool for other existing cell motility models. In this manner, we envision our model to be applied to different organisms, experimental setups, and artificial contour dynamics generated by different models, which is, however, beyond the scope of this paper. In our long-term quest for a quantitative, data-driven model of amoeboid motility, we want to use commonly-used biomarkers associated with the membrane and the cell cortex as direct inputs in our model and evaluate the predicted contour dynamics with experimental measurements. This way, the model acts as a computational bridge between the driving forces of cell motility, related to the F-actin density, and the resulting translocation of the cell membrane, described by the local motion of the cell contour.

The second main goal of our model is to simulate a variety of quantitatively different and realistic contour dynamics producing cell tracks that visually resemble experimental data very closely. Based on the model's capability to simulate such versatile contour dynamics, we deem it to be applicable to experimental cell tracks for varying degrees of motility and persistence, or even different types of locomotion. Since the majority of cell motility data rely on two-dimensional microscopy images, we have also chosen a 2D modeling approach. The theoretical framework of a 2D model, including the parametrization and propagation of the cell

membrane, is much easier to handle and, intuitively, better to understand than in 3D. Note that an extension of our approach to a 3D model is not straightforward and requires substantial conceptual improvements. A solid theoretical foundation of the 2D case can also serve as a starting point for the extension to 3D. Finally, a 2D model is advantageous due to its computational costs and feasibility.

Briefly, our model evolves two-dimensional contours representing the cell membrane and is based on three components. The first component is driven by a stochastic term generating membrane protrusions and is modeled by (1) the intensity of a self-exciting Poisson point process (so-called Hawkes process [75–77]) or, alternatively, (2) an Ornstein-Uhlenbeck like diffusion process. In this context, we show that the Hawkes process, due to its self-exciting nature, is suitable to produce a cascade of protrusion events to ensure a persistent cell migration. Key motility characteristics, i.e., duration, size, and number of these noise-induced protrusions, can be separately adjusted with different parameters, incorporating the individual cell behavior as well as differences in the extracellular environment. The second two components are mathematically well-defined geometric flows initiating contour retractions: the area-preserving curve-shortening flow (APCSF) to regularize the shape/arc length of the contour; and a further flow introduced as area adjustment flow (AAF) which expands/shrinks the cell contour with respect to a specified reference area. For more information on the APCSF, see [78–80]. Based on the interplay of the above components, the model defines the formation of protrusions as well as retractions. It is therefore linked to other mechanistic models evolving the cell contour as an elastic object in time and space [70].

In the following, we demonstrate how our model can be used to simulate realistic cell tracks. Then, we analyze experimental cell tracks (*D. discoideum*) by inferring the model-based protrusion and retraction components for two different locomotion types: the amoeboid and a so-called fan-shaped type, see [32, 81, 82] for more details. The inferred protrusion component is, then, compared to common biomarkers: the density of filamentous actin close to the membrane and its local motion. An implementation of the model, as well as a graphical user interface to simulate cell tracks, is provided in our Python-based toolbox `AmoePy` [83].

## Methods

### General notations and underlying coordinate system

The notation and theoretical framework used in this work have been established in [84]. Primarily, this framework is used as an analysis tool of amoeboid cell motility enabling us (1) to obtain smooth contour representations from stacks of segmented microscopy images, (2) to track quantities of interest along reference points (virtual markers) between successive contours to generate so-called kymograph plots, and (3) to identify protrusion and retraction events in an automated way. The framework was applied to artificial contour dynamics as well as cell tracks of different organisms and experimental setups. The first two aspects mentioned above, i.e., the process of generating smooth contour representations and the mapping of consecutive contours in time and space, are now used as part of our novel computational model to simulate contour dynamics and to analyze experimental data by inferring key motility characteristics based on this model. For a better understanding of our model, the framework presented in [84] is summarized in this section.

First, we demonstrated that smooth contours and the corresponding contour curvature can be derived easily from a discrete set of segmentation points (so-called active contour or snake) by using a Gaussian process regression (GPR) based on *a priori* covariance structure given by

some kernel. In our case, we used a Poisson kernel as underlying covariance function

$$k_r(\theta, \theta') = \frac{1 - r^2}{1 - 2r \cos(\theta - \theta') + r^2}, \quad \theta, \theta' \in [0, 2\pi), \ r \in [0, 1). \tag{1}$$

Briefly, the choice of $r_{\text{cont}}$ in $k_{r_{\text{cont}}}(\cdot, \cdot)$ affects the rigidity and stiffness of the resulting contour. Furthermore, an additional noise parameter $\sigma_{\text{noise}} > 0$ specifies the deviation between the initial segmentation points and the regression function obtained from the GPR with covariance structure:

$$\text{Cov}(\theta, \theta') = k_{r_{\text{cont}}}(\theta, \theta') + \sigma_{\text{noise}}^2 \delta(\theta - \theta'),$$

with $\delta(\cdot)$ denoting the Dirac delta function.

First, we consider $K \in \mathbb{N}$ cell contours denoted by $\Gamma_k$ with $k = 0, \ldots, K - 1$. The corresponding time points are denoted by $t_k = k \cdot \delta t$ with $\delta t > 0$. The coordinates of these contours are given by the following mapping

$$\Phi_k : [0, 2\pi) \to \mathbb{R}^2, \quad \theta \mapsto \Phi_k(\theta) = (\Phi_k^{(x)}(\theta), \Phi_k^{(y)}(\theta)), \tag{2}$$

where $\theta$ denotes the normalized arc length coordinate and $\Phi_k^{(x)}(\cdot)$ and $\Phi_k^{(y)}(\cdot)$ the x and y coordinates of the contour, respectively.

Noteworthy, connecting consecutive contours in time and space is intrinsically not well defined, i.e., there are multiple ways to do so [85–87]. In many approaches, varying constraints are introduced to track reference points/virtual markers from one contour to the next one, e.g., by using electrostatic field equations [85], level-set methods [85, 86], or mechanistic spring equations [87]. Naive contour mapping approaches include the propagation of virtual markers (VM) based on shortest paths to the next contour or by choosing paths in normal direction only. However, these approaches can change the order of neighboring virtual markers (so-called topological mapping violations). In [84], we address this issue by proposing a novel regularizing family of contour flows, connecting consecutive contours while preserving desirable characteristics of the underlying mapping trajectories.

For any flow between two contours $\Gamma_k$ and $\Gamma_{k+1}$, a mapping

$$\phi_k : [0, 2\pi) \to [0, 2\pi), \quad \theta \mapsto \phi_k(\theta)$$

is induced, which describes the translation along the contour under the flow. By assuming

$$\partial_\theta \phi_k(\theta) > 0,$$

we ensure that $\phi_k$ is a one-to-one mapping, i.e., mapping violations between $\Gamma_k$ and $\Gamma_{k+1}$ do not occur. Now, we can define a virtual marker trajectory starting at $\theta_0 \in [0, 2\pi)$ by the iteration

$$\theta_{k+1} = \phi_k(\theta_k). \tag{3}$$

In the following, we use a Lagrangian reference frame to describe geometric flows and the resulting evolution of virtual markers on the contour. This Lagrangian reference frame is denoted by $\chi_k$ and recursively defined by:

$$\chi_{k+1}(\theta_0) = \phi_k(\chi_k(\theta_0)), \quad \chi_0(\theta_0) = \theta_0.$$

For the limit of infinitely dense contours, we introduce the following notations

$$\Phi : [0, T] \times [0, 2\pi) \rightarrow \mathbb{R}^2 \qquad\qquad \chi : [0, T] \times [0, 2\pi) \rightarrow [0, 2\pi)$$
$$\text{and}$$
$$(t, \theta) \mapsto \Phi(t, \theta) \qquad\qquad\qquad (t, \theta) \mapsto \chi(t, \theta).$$

Given a VM $p_0 = \Phi_0(\tilde{\theta})$ on the first contour $\Gamma_0$ with arc length coordinate $\tilde{\theta} \in [0, 2\pi)$, we can now track this VM in time and space which corresponds to the function $t \mapsto \Phi(t, \tilde{\theta})$. Finally, we introduce a virtual marker distance ratio defined as:

$$\text{VMDR}_{t,\theta} = \partial_\theta \chi(t, \theta). \tag{4}$$

For the discrete case of $N \in \mathbb{N}$ virtual markers $\theta_{k,0}, \ldots, \theta_{k,N-1}$ on the contour $\Gamma_k$, this ratio can be rewritten as:

$$\text{VMDR}_{k,i} = \frac{|\theta_{k,i+1} - \theta_{k,i}|}{2\pi / N}, \tag{5}$$

measuring the distance between neighboring virtual markers divided by the distance of equidistantly spaced VMs.

In this discrete setting, by using regularized flows as in [84], a mapping $\phi_{k,\lambda_{\text{reg}}}$ between consecutive contours can be determined by solving the following optimization problem:

$$\phi_{k,\lambda_{\text{reg}}} = \underset{\phi_k}{\text{argmin}} \ F_k[\phi_k] + \lambda_{\text{reg}} \, U_k[\phi_k], \quad \lambda_{\text{reg}} \geq 0,$$

with cost functions $F_k[\phi_k]$ and $U_k[\phi_k]$ defined by:

$$F_k[\phi_k] \simeq \frac{1}{N\delta t^2} \sum_{i=0}^{N-1} \|\Phi_{k+1}(\theta_{k+1,i}) - \Phi_k(\theta_{k,i})\|^2,$$

$$U_k[\phi_k] \simeq N \sum_{i=0}^{N-1} |\theta_{k+1,i+1} - \theta_{k+1,i}|^2.$$

The first term $F_k$ is described by the sum over the lengths of all virtual marker trajectories from one contour to the consecutive contour, whereas the second term $U_k$ takes the sum over coordinate differences of neighboring virtual markers on the consecutive contour. Depending on a regularization parameter $\lambda_{\text{reg}} \geq 0$, this family of contour flows includes the two extreme cases: either enforcing shortest VM trajectories between contours ($\lambda_{\text{reg}} = 0$) or preserving equal distances between neighboring VMs for every time step ($\lambda_{\text{reg}} \gg 0$). For the continuous formulation of these regularized flows, see [84].

**Contour propagation vs. contour mapping.** We need to distinguish two different dynamics, which are both compatible with a given sequence of contours: (1) the contour propagation based on a model function $f : \mathbb{R}^+ \times \mathcal{S}^1 \rightarrow \mathbb{R}$ describing the normal velocity and (2) the "material" contour mapping under which each virtual marker is transported, leading to the following two-step algorithm. The dynamics of the contour itself are obtained by propagating an initially equidistant set of contour points for short time periods and with respect to the normal vector field

$$\left\langle \frac{\partial \Phi(t, \theta)}{\partial t}, \vec{n}(t, \theta) \right\rangle = f(t, \theta), \tag{6}$$

at time $t \in [0, T]$, normalized arc length coordinate $\theta \in [0, 2\pi)$, and $\langle \cdot, \cdot \rangle$ denoting the inner

product. A propagation in tangential direction affects the position of these contour points but does not affect the shape of the contour. Hence, the only information we have from the contour dynamics alone is the normal component of the actual material flow as expressed in Eq (6). However, such normal flow leads to thinning and clustering effects of the transported points over time. For this reason, we use a second kind of dynamics, namely regularizing flows described as in [84] to ensure an evenly-spaced distribution of virtual markers. This regularized flow is used to transport any dynamically relevant quantity over the dynamically evolving contours. Furthermore, the flow lines give rise to a coordinate system necessary to draw graphical representations (so-called kymographs) of each model component. In contrast to the contour propagation, the virtual markers under the regularizing flow are propagated also in tangential direction. In S1 Text, we present a detailed description of the model's implementation. In S1 Fig, we illustrate the two different kinds of marker trajectories: (1) the contour propagation (green dashed lines) and (2) the contour mapping (blue dashed lines) under which the stochastic protrusion process $X_{\text{prot}}(t, \theta)$ is transported and the underlying coordinate system is based on.

## Regularizing the shape and size of contours via geometric flows

Our model to evolve two-dimensional cell contours is based on three components: a protrusion term based on a stochastic process accounting for membrane protrusions and two geometric flows accounting for membrane retractions by regularizing the shape and area of the contour. Due to the separation into one protrusion component and two retraction components, the model provides an independent handling of these two key features of amoeboid cell motility.

The first geometric flow is defined by the area-preserving curve-shortening flow (APCSF) and denoted by $f_{\text{APCSF}}$. Briefly, the APCSF evolves a two-dimensional contour to a circle while preserving the area enclosed by the initial contour. It, therefore, minimizes the arc length of the contour without affecting the contour area. In the absence of other influences, the evolution of every contour to a circle is energetically favorable to reduce the surface tension of the membrane. For example, this behavior can be observed during cell death or by treating cells with Latrunculin to dissolve the actin cytoskeleton [88]. The second geometric flow, which we call area adjustment flow (AAF), is denoted by $f_{\text{AAF}}$. The AAF shrinks/expands the contour towards a predefined reference area. Since the underlying contour data relies on two-dimensional cross-sections of three-dimensional cells, the resulting contour area can change significantly. Therefore, a certain change of the contour area is desirable and possible in our model. By using both flows, we achieve a regularizing effect on the arc length (APCSF) and the area (AAF) necessary to counteract the forward movement initialized by the protrusion component $f_{\text{prot}}$. This component is based on a stochastic process, e.g., a Hawkes process or an Ornstein-Uhlenbeck process, and should be strictly positive since it describes the formation of protrusions only.

In Fig 1, simulated contour dynamics are shown based on: the protrusion component $f_{\text{prot}}$ only, $f_{\text{prot}}$ combined with $f_{\text{APCSF}}$, $f_{\text{prot}}$ combined with $f_{\text{AAF}}$, and a combination of all three components. In the first row (panel A), four different samples drawn from the same stochastic process $X_{\text{prot}}$ are displayed. The magnitude of this stochastic process is then directly translated into the protrusion component $f_{\text{prot}}$, see panel (B). The first two cases (panels (B) and C)) lead to significantly growing contours since the area adjustment is absent. In comparison, the contour dynamics in panel (C) are much smoother due to the regularizing effect of the APCSF. Due to the stochastic nature of our model, the occurrence of self-intersections during the evolution of the contour can not be ruled out. However, the APCSF eliminates noise-induced self-

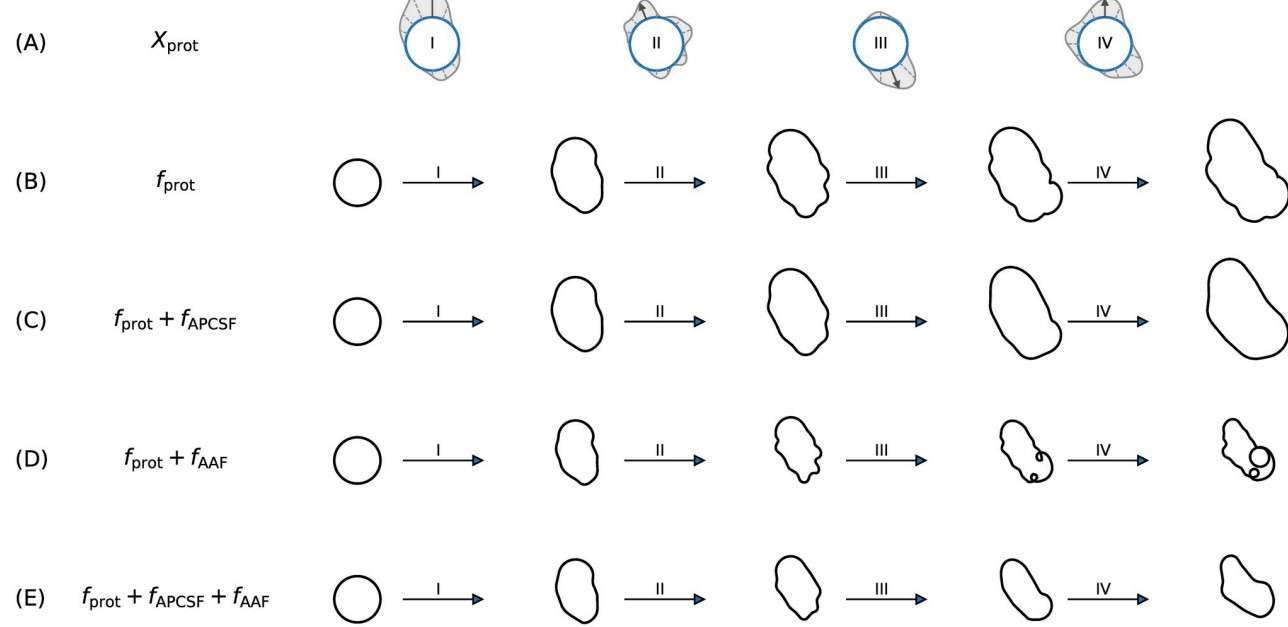

**Fig 1. Rationale why all three model components are necessary. (A)** Realizations of a stochastic process $X_{\text{prot}}$ driving the protrusion component $f_{\text{prot}}$.
**(B)** Simulated contour dynamics only based on protrusion component leading to significant contour growth. **(C)** By combining the protrusion
component with the APCSF, we obtain smoother and similarly sized contours. **(D)** On the contrary, combining the protrusion component with the
AAF only results in highly curved contours with possible self-intersections. **(E)** A combination of all three components is necessary to obtain stable
contour dynamics over time.

intersections over time. If the APCSF is absent, as shown in panel (D), dynamics of similarly
sized but also highly curved and possibly self-intersecting contours are produced. Therefore, a
combination of all three components, as in our model, is necessary (last row).

**Area-preserving curve-shortening flow.** To regularize the contour arc length in our
model, we used the APCSF:

$$\left\langle \frac{\partial \Phi(t, \theta)}{\partial t}, \vec{n}(t, \theta) \right\rangle = -\left( \kappa(t, \theta) - \frac{2\pi}{L(\Phi(t, \cdot))} \right), \tag{7}$$

where $\kappa(t, \theta)$ denotes the curvature and $\vec{n}(t, \theta)$ the outward-pointing normal vector at time
$t \in \mathbb{R}^+$ for a virtual marker with arc length coordinate $\theta \in [0, 2\pi)$ on the first contour. Here, $L$
$(\Phi(t, \cdot))$ denotes the total arc length and is defined by the following functional:

$$L(t) = L(\Phi(t, \cdot)) = \int_0^{2\pi} \left\| \frac{\partial \Phi(t, \theta)}{\partial \theta} \right\|_2 \, d\theta.$$

The APCSF is defined as the gradient flow of this functional under the area-preserving con-
straint $A(t) := A(\Phi(t, \cdot)) = A(0)$ for all $t > 0$, where the contour area is defined by:

$$A(t) = A(\Phi(t, \cdot)) := \int_0^{2\pi} \Phi^{(x)} \frac{\partial \Phi^{(y)}}{\partial \theta} \, d\theta = \int_0^{2\pi} \Phi^{(y)} \frac{\partial \Phi^{(x)}}{\partial \theta} \, d\theta.$$

As a consequence, the APCSF evolves every contour to a circle of the same area, minimizing
the contour arc length to $L(t) \xrightarrow{t \to \infty} 2\sqrt{\pi A(0)}$ while maintaining its area. Of note, under the
APCSF, a contour with self-intersections is also evolved into a circle. However, in this case, the

area of the contour is not well-defined and the area-preserving property of the APCSF should be viewed with caution.

**Area adjustment flow.**    While the APCSF has no effect on the contour area, the protrusion component would expand the cell contour most of the time and therefore also its area. The area adjustment flow counteracts this expansion. It is defined as

$$\left\langle \frac{\partial \Phi(t, \theta)}{\partial t}, \vec{n}(t, \theta) \right\rangle = -\frac{A(t) - A_{\text{ref}}}{A_{\text{ref}} \cdot L(t)} \left\langle \Phi(t, \theta) - \Phi_{\text{CM}}(t), \vec{n}(t, \theta) \right\rangle, \tag{8}$$

with reference area $A_{\text{ref}} \in \mathbb{R}^+$ and center of mass trajectory $\Phi_{\text{CM}} : \mathbb{R}^+ \rightarrow \mathbb{R}^2$ defined by

$$\Phi_{\text{CM}}(t) = \frac{1}{2\pi} \int_0^{2\pi} \Phi(t, \theta) d\theta.$$

As reference area $A_{\text{ref}}$, we choose the 1st percentile of the entire area time series, individually, for each experimental cell track.

It is easy to see that an area adjustment is achieved by this flow. Furthermore, the AAF affects the contour in normal direction only and is shape-preserving if used *without* the other two components. In S1 Text, we introduce another area regularizing flow which is based on the gradient flow of the area functional. This flow minimizes the contour area much more rapidly by affecting the contour in normal direction only. However, this flow is not shape-preserving.

## Intensity of a Hawkes process as protrusion component

Statistical analyses of different sequences of cell contours have shown that a protrusion event increases the probability of nearby follow-up protrusions [17]. We, therefore, modeled the protrusion component $f_{\text{prot}}$ in our model by the intensity of a self-exciting Poisson point process, a so-called Hawkes process. Due to the self-exciting nature of the Hawkes process, a cascade of protrusion events can be generated which results in substantial and persistent contour changes. In this way, we obtain contour dynamics with a significantly moving center of mass instead of a fluctuating membrane only.

The intensity $\lambda(t, \theta)$ of a spatio-temporal Hawkes process is defined by

$$\lambda(t, \theta) = \mu(\theta) + \sum_{i : t_i < t} g(t - t_i, \theta - \theta_i), \tag{9}$$

with event times $\{t_1, t_2, \ldots\}$, background intensity function $\mu : [0, 2\pi) \rightarrow \mathbb{R}^+$ and kernel function $g : [0, T] \times [0, 2\pi) \rightarrow \mathbb{R}^+$, characterizing the positive influence of past events (ancestors) on the emergence of future events (descendants). The background intensity is defined in terms of the normalized Poisson kernel $\tilde{k}_{r_{\text{pol}}}$:

$$\tilde{k}_r(\theta, \theta') = \frac{k_r(\theta, \theta')}{\sqrt{k_r(\theta, \theta) k_r(\theta', \theta')}} = \frac{(1 - r)^2}{1 - 2r \cos(\theta - \theta') + r^2}, \tag{10}$$

with $\theta, \theta' \in [0, 2\pi)$ and $r \in [0, 1)$. The normalized Poisson kernel holds the following properties: $\left(\frac{1-r}{1+r}\right)^2 \leq \tilde{k}_r(\theta, \theta') < 1$ for all $\theta \neq \theta'$ and $\tilde{k}_r(\theta, \theta') = 1$ if and only if $\theta = \theta'$. In S2 Fig, the Poisson kernel function from Eq (1) and its normalized version from Eq (10) are shown for different parameters $r \in [0, 1)$. As background intensity function, we now define

$$\mu(\theta) = \lambda_0 \frac{\tilde{k}_{r_{\text{pol}}}(\theta, \pi)}{\int_0^{2\pi} \tilde{k}_{r_{\text{pol}}}(\theta, \pi) \, d\theta}, \tag{11}$$

with background rate $\lambda_0 > 0$ and $\tilde{k}_{r_{\text{pol}}}$ as in Eq (10) with $0 \leq r_{\text{pol}} < 1$. For the non-polarized case $r_{\text{pol}} = 0$, the background intensity simplifies to $\mu = \frac{\lambda_0}{2\pi}$. For $r_{\text{pol}} > 0$, polarization takes place with a local maximum at $\theta = \pi$, and local minima at $\theta = 0$ and $\theta \to 2\pi$. For the sake of simplicity, the polarization is fixed for the entire time span and also with respect to the local position on the contour ($\theta' = \pi$). One could add a time dependence to the background intensity with a changing cell front, i.e., with a polarization angle $\theta'$ in Eq (10) varying over time and depending on the proximity of external cues, repelling contours, or obstacles. In this case, however, the Hawkes process needs to be realized sequentially over time while evolving the cell contour. In contrast, in our test case with a locally fixed and time-independent polarization, the Hawkes process can be realized in advance before the contour is propagated.

We used a product kernel function $g(t, \theta) = g_1(t) \cdot g_2(\theta)$ with temporal component $g_1(t)$ and spatial component $g_2(\theta)$. The temporal kernel function is given by

$$g_1(t) = \alpha \beta \, t \, e^{-\beta t},$$

with arrival intensity $\alpha > 0$ and exponential decay rate $\beta > 0$. As spatial kernel, we used the von Mises distribution,

$$g_2(\theta) = \frac{e^{\kappa_M \cos(\theta)}}{2\pi I_0(\kappa_M)},$$

with $\kappa_M > 0$ as concentration parameter and $I_0(\kappa_M)$ denoting the modified Bessel function of order 0.

For a realization of the Hawkes process, the protrusion process $X_{\text{prot}} : \mathbb{R}^+ \times \mathcal{S}^1 \to \mathbb{R}$ is defined as:

$$X_{\text{prot}}(t, \theta) = \frac{c_s}{\text{VMDR}(t, \theta)} \sum_{i:t_i<t} \tilde{g}(t - t_i, \theta - \theta_i), \tag{12}$$

with any spatio-temporal kernel function $\tilde{g} : [0, T] \times [0, 2\pi] \to \mathbb{R}^+$, time scaling factor $c_s > 0$, and VMDR as in Eq (4) accounting for local contour/arc length changes. For the sake of simplicity, we choose $c_s = 1s$ and the same kernel function as above, i.e., $\tilde{g}(t, \theta) \equiv g(t, \theta)$. Hence, we can rewrite the protrusion process $X_{\text{prot}}$ in terms of the intensity $\lambda(t, \theta)$ of a Hawkes process realization, defined as in Eq (9):

$$X_{\text{prot}}(t, \theta) = \frac{c_s}{\text{VMDR}(t, \theta)} (\lambda(t, \theta) - \mu(\theta)), \tag{13}$$

to highlight that $X_{\text{prot}}$ directly resembles the intensity of the Hawkes process and not the underlying realization of point events or the Hawkes process itself.

Based on the underlying choice of parameters, different motility characteristics can be adjusted with our model:

- the number of protrusions by $\lambda_0$ and $\alpha$,

- the duration of protrusions by $\beta$,

- the size of protrusions (many small protrusions vs. a single large protrusion) by $\kappa_M$,

- membrane fluctuation vs. creation of pseudopods with a substantial movement of the center of mass by $\alpha$,

- non-polarized vs. polarized contour dynamics by $r_{\text{pol}}$.

Furthermore, the velocity of contour changes during the creation of protrusions and retractions can be adjusted with additional model weights which are introduced in the following section. Of note, the parameter regime of our model incorporates both, the individual cell behavior as well as differences in the extracellular environment.

In S3 Fig, the kernel functions $g_1(t)$ and $g_2(\theta)$ are illustrated for varying parameters $\alpha$, $\beta$, and $\kappa_M$ as well as the Hawkes intensity $\lambda(t, \theta)$ for a fixed set of parameters. In S1 Text, we illustrate an alternative approach by modeling the protrusion component with an Ornstein-Uhlenbeck type of diffusion process.

## Three-component contour dynamics model

In this section, we formulate our contour dynamics model based on the three components: a protrusion component based on a stochastic process $X_{\mathrm{prot}}$, e.g., a Hawkes process defined as in Eq (13), the APCSF from Eq (7), and the AAF from Eq (8). In our model, the following parameters are required:

- weight parameters $w_{\mathrm{prot}}$, $w_{\mathrm{APCSF}}$, $w_{\mathrm{AAF}} > 0$,

- a reference area $A_{\mathrm{ref}} > 0$,

- additional parameters regarding the stochastic process $X_{\mathrm{prot}}$.

Furthermore, the computation of the following geometric quantities is necessary:

- contour area $A(t) \in \mathbb{R}^+$ and arc length $L(t) \in \mathbb{R}^+$,

- contour curvature $\kappa(t, \theta) \in \mathbb{R}$,

- center of mass trajectory $\Phi_{\mathrm{CM}}(t) \in \mathbb{R}^2$,

- outward-pointing unit normal vectors $\vec{n}(t, \theta) \in \mathbb{R}^2$.

For a virtual marker $\Phi(0, \theta)$ with initial arc length coordinate $\theta \in [0, 2\pi)$, the normal component of its trajectory $t \mapsto \Phi(t, \theta)$ with $t \in [0, T]$ is given by

$$\left\langle \frac{\partial \Phi(t, \theta)}{\partial t}, \vec{n}(t, \theta) \right\rangle = f(t, \theta), \tag{14}$$

where $f : \mathbb{R}^+ \times \mathcal{S}^1 \to \mathbb{R}$ is defined as

$$f = f_{\mathrm{prot}} + f_{\mathrm{APCSF}} + f_{\mathrm{AAF}}, \tag{15}$$

with the following three components:

$$
\begin{aligned}
I: \quad & f_{\mathrm{prot}}(t, \theta) && = w_{\mathrm{prot}} \frac{X_{\mathrm{prot}}(t, \theta)}{L(t)}, \\
II: \quad & f_{\mathrm{APCSF}}(t, \theta) && = -w_{\mathrm{APCSF}} \left( \kappa(t, \theta) - \frac{2\pi}{L(t)} \right), \\
III: \quad & f_{\mathrm{AAF}}(t, \theta) && = -w_{\mathrm{AAF}} \frac{A(t) - A_{\mathrm{ref}}}{A_{\mathrm{ref}} \cdot L(t)} \left\langle \Phi(t, \theta) - \Phi_{\mathrm{CM}}(t), \vec{n}(t, \theta) \right\rangle.
\end{aligned}
\tag{16}
$$

Based on the first equation, the protrusion component $f_{\mathrm{prot}}$ is mainly affected by $X_{\mathrm{prot}}$ for which we can insert different stochastic processes, e.g., a Hawkes process, an Ornstein-Uhlenbeck process, or a Poisson point process.

First, in Fig 1, we have shown the interplay of the three model components and why all three components are required to produce stable and realistic contour dynamics. Now,

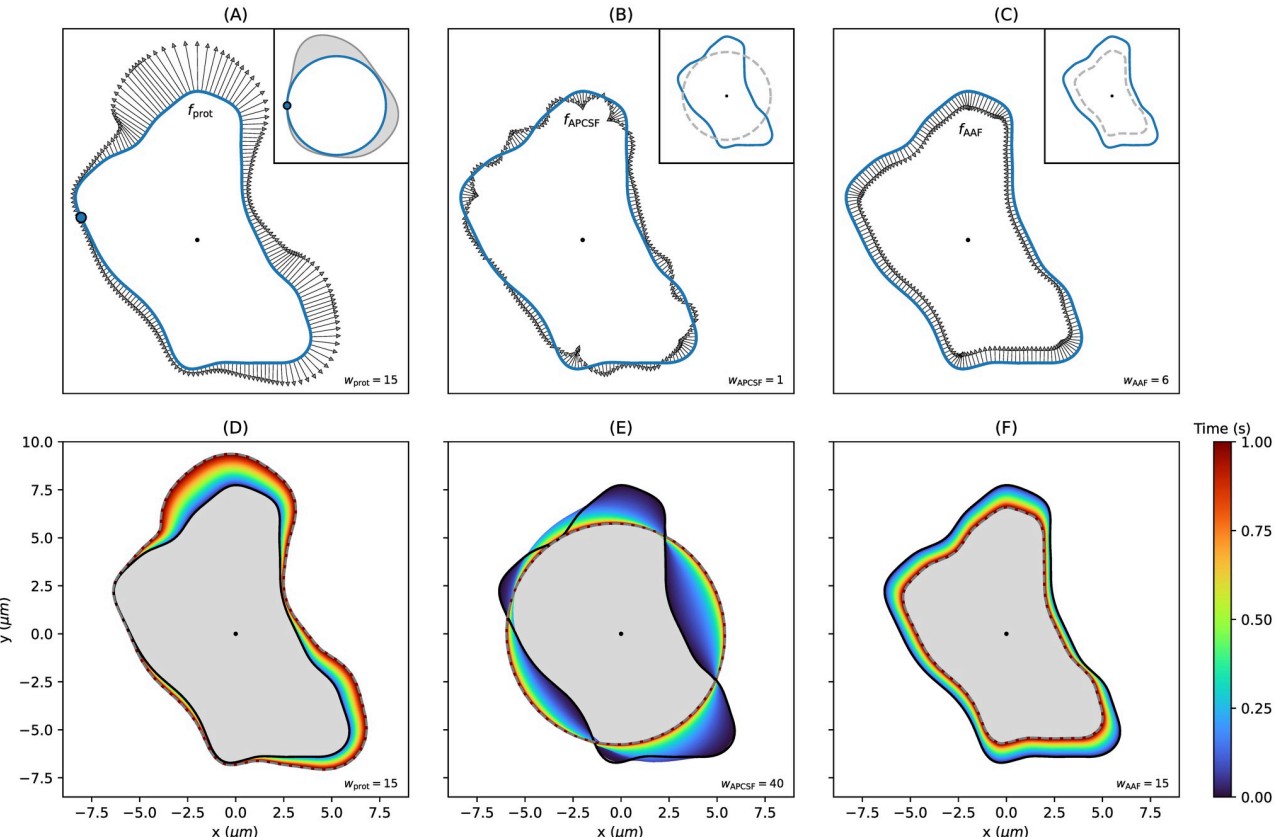

**Fig 2. Comparison of the three model components: $f_{\text{prot}}$, $f_{\text{APCSF}}$, and, $f_{\text{AAF}}$. (A)** Protrusion component driven by a stochastic process (generated sample shown in top right corner). The reference point $\theta = 0$ is highlighted as blue dot. **(B, C)** APCSF and AAF with final states displayed in corresponding top right corners. **(D-F)** Example contour propagated individually by only one component for a time span of 1$s$. Regarding the contour parametrization, the reference area is given by $A_{\text{ref}} = 60\mu m^2$ with GPR hyperparameters $r_{\text{cont}} = 0.4$ and $\sigma_{\text{noise}} = 0.01$.

separately generated contour dynamics, each being based on one model component only, are shown in Fig 2. In panel (A), the protrusion component is displayed with a generated sample of the underlying stochastic process $X_{\text{prot}}$ in the top right square. In this example, $X_{\text{prot}}$ is defined as correlated noise on a unit circle and is mapped onto the contour with respect to the contour arc length and a reference point $\theta = 0$ (blue dot). In panel (B), the APCSF is shown with its final state (dashed grey contour): a circle with the same area as the initial contour. Since the APCSF is mainly defined by the contour curvature, the contour shrinks faster for regions with large positive (convex) curvature and expands faster for regions with large negative (concave) curvature. In panel (C), it can be observed that the area adjustment is weaker for contour segments that are closer to the center of mass. The final state is again displayed as a dashed gray contour.

Finally, in panels (D-F), we display test scenarios of simulated contour dynamics each based on a single component only. In panel (D), we observe a steadily growing contour since the $f_{\text{prot}}$ is the only active component. Under the APCSF (panel (E)) the contour evolves into a circle by expanding concave parts and shrinking convex parts of the contour. In the case of AAF being the only active component, we observe a shape-preserving shrinkage of the contour. In this case, the reference area ($A_{\text{ref}} = 60\mu m^2$) was set to be smaller than the initial contour area. Alternatively, by choosing a larger reference area, a shape-preserving growth of the

contour would occur. The contours in panels (D-F) are propagated in normal direction only without any influence of VM remapping/regularizing flows.

## Inference of model components and parameter estimation

Besides simulating contour dynamics, our model is used to infer the different model components $f_{\text{prot}}, f_{\text{APCSF}}, f_{\text{AAF}}$ of experimental cell tracks to analyze and characterize cell tracks on the individual level. Often, the analysis of experimental contour dynamics is based on the cell's local motion, i.e., the velocity of the contour. However, the local motion comprises characteristics of the entire contour dynamics: protrusions, retractions, as well as minor membrane fluctuations. In our approach, by inferring the different model components, the protrusions and retractions can be quantified separately from each other. Furthermore, by estimating the corresponding component weights, different motility types, i.e., the amoeboid type and the fan-shaped type, can be classified.

Since the APCSF and AAF from Eq (16) are based on the contour curvature, arc length, and area; $f_{\text{APCSF}}$ and $f_{\text{AAF}}$ can be computed separately for each time step solely based on the current contour. For this reason, one can explicitly determine the only remaining component $f_{\text{prot}}$ to propagate one contour to the next one. This approach enables us to replicate the experimental cell track with our model for an estimated set of model parameters. This set includes the reference area $A_{\text{ref}}$ and three model component weights $w_{\text{prot}}, w_{\text{APCSF}},$ and $w_{\text{AAF}}$.

While the APCSF does not affect the contour area, the (positive) protrusion component increases the contour area. Therefore, the reference area $A_{\text{ref}}$ should be chosen relatively small to achieve a counteracting shrinkage effect of the AAF. In our case, we have chosen the 1st percentile of the entire area time series of the cell track. Next, we estimate $f_{\text{APCSF}}$ and $f_{\text{AAF}}$ and their corresponding weights, $w_{\text{APCSF}}$ and $w_{\text{AAF}}$, by making use of the local motion kymograph of the underlying cell track. More precisely, we determine $w_{\text{APCSF}}$ and $w_{\text{AAF}}$ such that negative regions (i.e. retractions) of the local motion are optimally captured by the above components. Initially, the remaining component $f_{\text{prot}}$ is estimated by deducting $f_{\text{APCSF}}$ and $f_{\text{AAF}}$ from the local motion kymograph. Afterwards, we choose $w_{\text{prot}}$ such that the sample variance of the underlying protrusion process $X_{\text{prot}}$ is standardized with $\text{Var}(X_{\text{prot}}) = 1$. In the next step, we further tune the inferred protrusion component $f_{\text{prot}}$ by minimizing the distances of virtual markers propagated with respect to $f_{\text{prot}}$ and the "receiving" contour at the next time step. In this optimization step, we use the built-in least-squares method from the Python package `SciPy`.

Finally, we evaluate the goodness of fit of the inferred protrusion component. Since $f_{\text{APCSF}}$ and $f_{\text{AAF}}$ are computed in advance, the remaining protrusion component corrects any missing contour dynamics to propagate one contour to the next one. For this reason, the inferred protrusion component can be negative if a contour retraction is stronger than the APCSF and the AAF have predicted. Therefore, a good fit is given if the estimated protrusion component is (mostly) positive, i.e., the contour retractions are successfully captured by the other two components.

## Results

### Hawkes process based simulations of amoeboid cell motility

In this section, we show that the Hawkes process is suitable for simulating ameboid cell motility. With the proposed model a variety of qualitatively different contour dynamics were generated.

The protrusion component in this section is based on a Hawkes process defined as in Eq (13). The underlying model weights were set to $w_{\text{prot}} = 7.5 \mu m^2/s$, $w_{\text{APCSF}} = 0.1 \mu m^2/s$, and

$w_{\text{AAF}} = 1\mu m/s$. As reference area, we have chosen $A_{\text{ref}} = 80\mu m^2$. First, we simulated contour dynamics based on a non-polarized test scenario realized by choosing $r_{\text{pol}} = 0$. Then, we have chosen a polarized test scenario by choosing $r_{\text{pol}} = 0.5$. A summary of all parameter values is displayed in Table 1. As initial contour, we have chosen a circle with an area equal to $A_{\text{ref}}$.

Fig 3 shows exemplary non-polarized contour dynamics, which are driven by a Hawkes process. We observed that the Hawkes process can enforce a substantial change of the center of mass trajectory as displayed in panels (A) and (B). The self-excitation can be also observed in the kymograph of the protrusion component $f_{\text{prot}}$ (panel (D)). In this kymograph, point events realized from the Hawkes process are depicted as circles and show clusters as expected from the self-excitation property. The corresponding Hawkes intensity was then used to define the protrusion process in our contour dynamics model as in Eq (13). The same red patterns of the $f_{\text{prot}}$ kymograph can be also found in the final local motion kymograph (panel (C)). The retractions of the contour dynamics are mainly defined by the area adjustment $f_{\text{AAF}}$ (panel (E)). By comparing this kymograph with the area plot in panel (G), we see that dark blue patterns, indicating strong retractions, occur when the contour area $A(t)$ is much larger than the reference area $A_{\text{ref}}$.

Due to the definition of the APCSF component $f_{\text{APCSF}}$, the curvature at each part of the contour can be inferred from the corresponding kymograph (panel (F)). Since the contour dynamics start from a perfect circle, possessing a constant and relatively small curvature, the kymograph starts with values close to zero. Later on, blue horizontal stripes indicate the position of protrusions and the rear of the cell, which possess a large positive curvature and are retracted therefore by the APCSF. In contrast, concave regions of the cell contour correspond to red horizontal stripes since they are enforced to expand under the APCSF. Finally, the influence of the APCSF, minimizing the contour arc length, is shown in the plot in panel (H).

Fig 4 shows an artificial cell track based on a polarized Hawkes process. In contrast to the cell track of Fig 3, the contour dynamics are characterized by higher motility and stronger persistence. Interestingly, we observed the zigzag pattern well known from experimental amoeboid cell tracks [17]. The self-exciting behavior of the Hawkes process initialized with a

**Table 1. Choice of parameters and meaning.**

| Parameter | Value | Unit | Meaning |
|---|---|---|---|
| **Contour parametrization** | | | |
| $r_{\text{cont}}$ | 0.6 | – | GPR smoothing |
| $\sigma_{\text{noise}}$ | 0.05 | – | GPR noise |
| $\lambda_{\text{reg}}$ | 10 | $\frac{\mu m^2}{s^2}$ | Flow regularization |
| $A_{\text{ref}}$ | 80 | $\mu m^2$ | Reference area |
| **Hawkes process** | | | |
| $\lambda_0$ | 1 | $s^{-1}$ | Background intensity |
| $\alpha$ | 0.4 | $s^{-1}$ | Arrival intensity |
| $\beta$ | 0.5 | $s^{-1}$ | Exponential decay rate |
| $\kappa_M$ | 100 | – | Spatial concentration |
| $r_{\text{pol}}$ | 0 and 0.5 | – | Polarization |
| **Model weights** | | | |
| $w_{\text{prot}}$ | 7.5 | $\frac{\mu m^2}{s}$ | Protrusion weight |
| $w_{\text{APCSF}}$ | 0.1 | $\frac{\mu m^2}{s}$ | APCSF weight |
| $w_{\text{AAF}}$ | 1 | $\frac{\mu m}{s}$ | AAF weight |

List of parameters for simulated contour dynamics based on a Hawkes process.

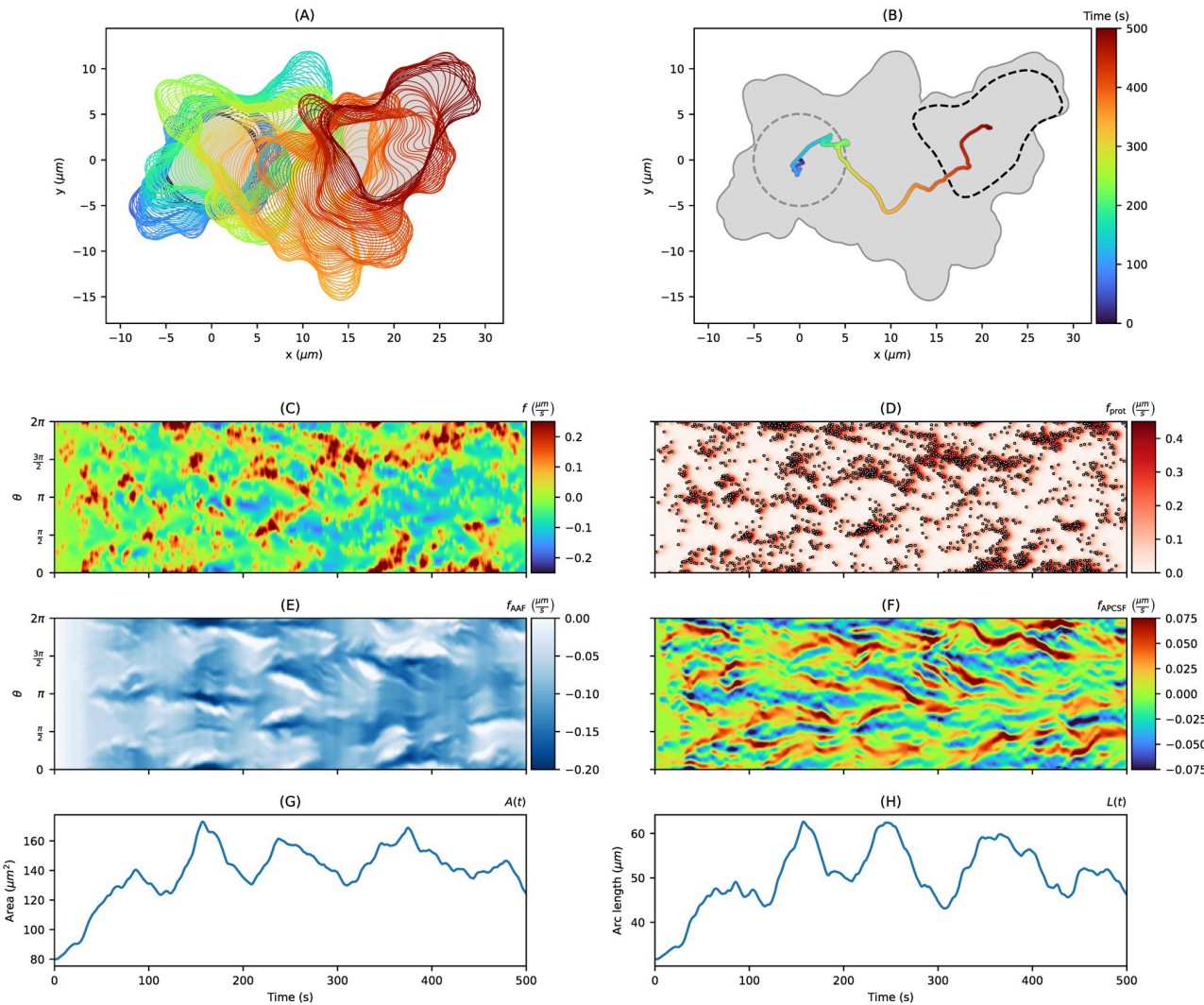

**Fig 3. Artificial non-polarized cell track with $T = 500s$ based on a Hawkes process. (A, B)** Contour dynamics (left, colored contours), the center of mass trajectory (right, colored line), and the trace of the entire cell track(right, gray area). **(C, D)** In the second row, kymographs of the local motion $f$ and its protrusion component $f_{\text{prot}}$ are displayed. Point events realized by the underlying Hawkes process are depicted as circles in the protrusion kymograph. **(E, F)** In the third row, kymographs of the other two components $f_{\text{AAF}}$ and $f_{\text{APCSF}}$ are shown. Finally, in panels **(G, H)**, the evolution of the contour area as well as the contour arc length are presented.

spatially unimodal background intensity (see S2 Fig) is sufficient to reproduce the zigzag movement. For this case, the clustering of point events (panel (D)) is even more pronounced than for the non-polarized case. From the local motion kymograph, we infer that the protrusions (red regions) occur mainly at $\theta = \pi$ defining the front of the cell. On the other side, retractions (blue regions) occur near $\theta = 0$ and $\theta = 2\pi$. In contrast to the non-polarized scenario, the curvature lines in the kymograph representing $f_{\text{APCSF}}$ are less horizontal. Instead, the characteristic diagonal stripes stand for protrusions created at the front of the cell which are then moved along both sides of the contour until they reach the rear of the cell. This has also been observed experimentally in [69].

In addition to the Hawkes process, we also modeled the creation of protrusions by a simple Poisson point process. We generated cell tracks for each of the two scenarios described above

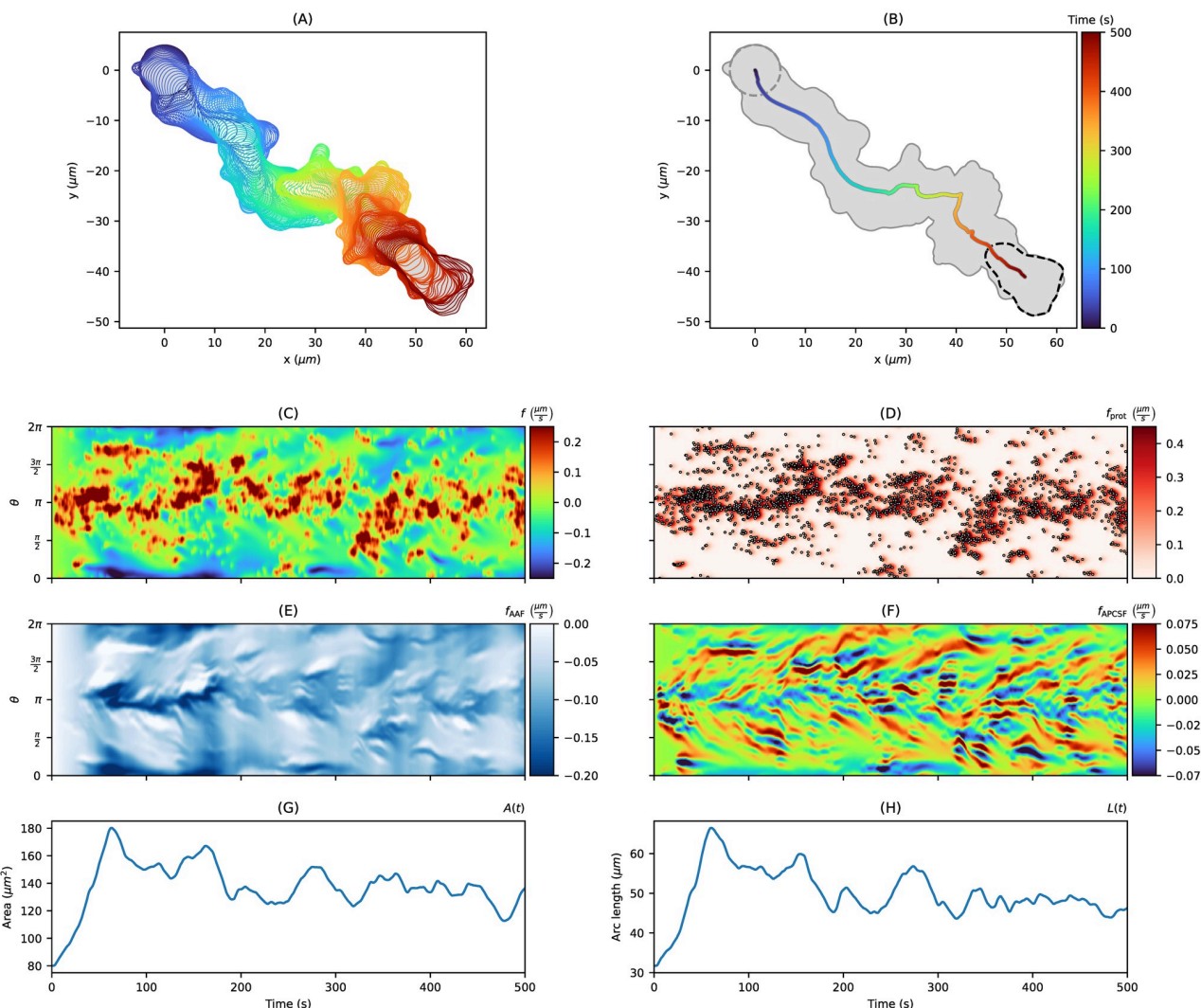

**Fig 4. Artificial polarized cell track with $T = 500s$ based on a Hawkes process.** The contour dynamics, the center of mass trajectory, the contour trace, the kymographs of $f$, $f_{prot}$, $f_{APCSF}$, and $f_{AAF}$ as well as the contour area and arc length are shown in the same order as in Fig 3.

where the protrusion component is only defined by the first generation of point events without further offspring as for the Hawkes process. To achieve the same number of point events, we increased the background intensity from $\lambda_0 = 1s^{-1}$ to $\lambda_0 = 5s^{-1}$.

In S4 and S5 Figs, five cell tracks driven by such a standard (i.e. not self-exciting) Poisson process are displayed, respectively. In comparison to the cell tracks generated by the Hawkes process, we observe a more equal distribution of point events with less significant event clusters. Therefore, for the non-polarized scenario, membrane fluctuations without large displacements of the center of mass were observed. For the polarized scenario, the cell moves persistently in one direction without having major turns, zigzag movements, or additional *de novo* pseudopodia. Hence, the Hawkes process was better suited to model amoeboid cell motility than a standard Poisson point process.

Animations of the contour dynamics and the corresponding kymographs from Figs 3 and 4 are shown in S1 and S2 Videos, respectively. In these videos, each point event generated by the

Hawkes process is illustrated as a circle in the protrusion component kymograph (second row) and the animated contour dynamics (bottom left). The effect of the self-excitation of the Hawkes process can be clearly seen in form of cascades of point events determining the direction and the movement of the cell.

We furthermore examined a second alternative to the Hawkes process, an Ornstein-Uhlenbeck type process, see S1 Text for more details. However, the resulting simulations are less satisfactory due to a higher number of protrusive features that are not observed in experiments and additional artifacts such as a pulsating membrane and a swimming type of locomotion. Since a self-exciting property is missing in this approach, it is much more difficult to generate cascades of multiple protrusions and subsequent reorientation phases of the contour dynamics. Contour dynamics and the corresponding model components obtained from this approach are displayed in S1 Text with animations shown in S3 Video.

**How to choose the regularization parameter $\lambda_{reg}$?.**     To ensure stable contour dynamics within our model, we used regularized flows defined as in [84]. Based on a regularization parameter $\lambda_{reg} \geq 0$, we can enforce a more even distribution of virtual markers for every time step/contour. This way, thinning/clustering effects of virtual markers over time can be avoided, see S1 Text and S1 Fig for more details.

In this section, we investigated the influence of this regularization on the overall contour dynamics and studied, under which circumstances, clustering/thinning effects of virtual markers as well as other artifacts can occur. Moreover, we challenged our model by varying the temporal resolution. In this case, the underlying contour mapping is also affected since the number of contours is increased/reduced.

In Fig 5, we present simulations of non-polarized cell tracks (panel (B)) as well as polarized cell tracks (panel (C)) generated with the parameter choices in Table 1 but for varying regularization parameter $\lambda_{reg} \in \{0.01, 0.1, 10, 1000\}$. The trace of each cell track (gray area) and the corresponding centroid trajectories (colored lines) are displayed. For the non-polarized case (panel (B)), the overall contour dynamics were substantially altered in cases of weak ($\lambda_{reg} = 0.01$) or strong regularization ($\lambda_{reg} = 1000$). For a medium regularization ($\lambda_{reg} = 0.1, 10$), the model produced contour dynamics similar to each other with almost identical contours at the end of each track. Analogous observations were made for the polarized test cases in panel (C). By decreasing $\lambda_{reg}$, the overall motility is also reduced. This can be understood from Eq (13): a low regularization results in thinning effects of VMs at the leading edge which increases the virtual marker distance ratio (VMDR) and therefore decreases the protrusion process. In panel (C), we noticed differences in the main direction of the cell track due to different contour mappings at the beginning of each track. However, the overall contour dynamics were not affected by increasing $\lambda_{reg}$.

In panels (D) and (E), histograms are displayed showing the effect of each regularization scheme on the distribution of virtual markers for non-polarized (left) and polarized (right) contour dynamics. Each histogram displays the relative frequencies with respect to the virtual marker distance ratio from Eq (5) for all contours of a simulated cell track. In case of a weak regularization, the thinning and clustering effect are reflected by a higher variance of the VMDR with distances up to 2 times larger than in the equidistant case. Especially in panel (E) for $\lambda_{reg} = 0.01$ (light blue histogram), a major peak at VMDR $\approx 0.5$ indicates a strong clustering of virtual markers at the rear of the cell. In the case of a strong regularization, an almost even distribution of virtual markers was achieved for every time step/contour which is reflected by VM distance ratios close to 1.

In panels (F) and (G), kymographs of the APCSF component corresponding to the non-polarized (left) and polarized (right) tracks are displayed. For a weak regularization ($\lambda_{reg} = 0.01$), prominent thinning and clustering effects of virtual markers were observed, see dashed

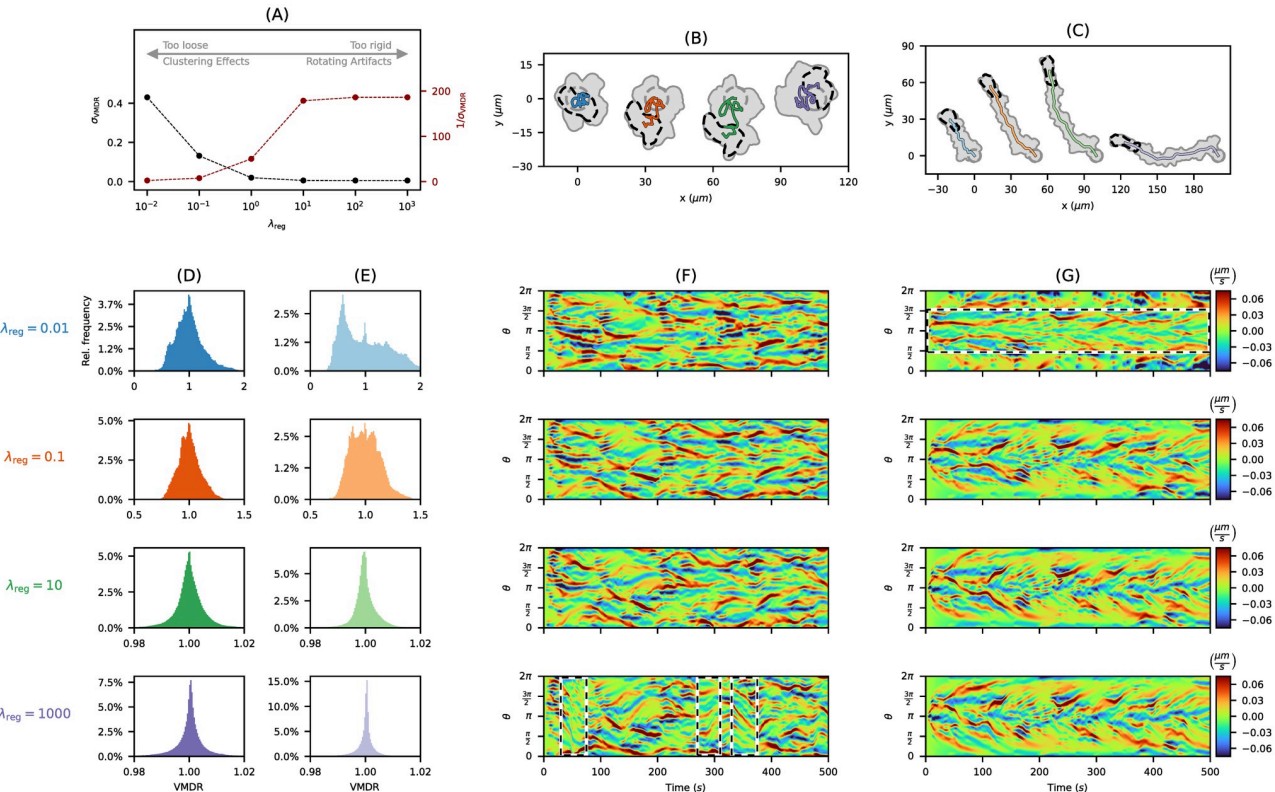

**Fig 5. Influence of regularization parameter $\lambda_{reg}$ on simulated cell tracks. (A)** The impact of $\lambda_{reg}$ on the virtual marker distribution described by the standard deviation of the virtual marker distance ratio $\sigma_{VMDR}$ and its reciprocal. **(B, C)** Non-polarized (left) and polarized (right) cell tracks generated with varying regularization parameter $\lambda_{reg} \in \{0.01, 0.1, 10, 1000\}$. The same protrusion component is underlying in **(B)** and **(C)**, respectively. **(D, E)** Histograms of the VMDR for varying $\lambda_{reg}$ corresponding to the non-polarized (left) and polarized (right) cell tracks from panels (B) and (C). **(F, G)** Kymographs of the APCSF component of all non-polarized (left) and polarized (right) cell tracks with regions of interest shown as black and white dashed boxes.

box in panel (G). In the case of a strong regularization ($\lambda_{reg}$ = 1000), an equidistant distribution of virtual markers is enforced at each time step. As a consequence, an emerging protrusion has a direct effect on all virtual markers along the cell contour leading to a cyclic shift of all markers. These shifts were observed as rotating elements of the cell contour indicated by sharp diagonals in the APCSF kymograph, see dashed boxes in panel (F). For medium regularization choices ($\lambda_{reg}$ = 0.1, 10), the above artifacts (clustering effects, rotating elements) were not observed.

A summary of our findings is presented in Fig 5(A), where the influence of $\lambda_{reg}$ on the distribution of virtual markers is shown. In this context, we computed the standard deviation $\sigma_{VMDR}$ of the virtual marker distance ratio for varying $\lambda_{reg}$. For a weak regularization $\lambda_{reg}$ = 0.01, a loose connection of neighboring virtual markers is observed which leads to thinning and clustering effects. If the regularization is chosen too strong ($\lambda_{reg} = 10^2, 10^3$), other artifacts such as rotating elements of the contour might occur. For this reason, we recommend a medium regularization: $0.1 \leq \lambda_{reg} \leq 10$. In our simulations, we have chosen the upper limit $\lambda_{reg}$ = 10 to obtain a more equidistant distribution of virtual markers which ensures stable contour dynamics even for a longer time period $T > 500s$ while avoiding rotating artifacts of the contour.

For more details, see S6 Fig, where the above non-polarized cell tracks are shown for a wider selection of varying regularization parameters $\lambda_{reg} \in \{10^{-2}, 10^{-1}, \ldots, 10^3\}$ and with all

corresponding model components. Again, for the intermediate cases $\lambda_{\text{reg}} \in [0.1, 10]$, we noticed only very few differences in all shown kymographs. However, because of small changes of the contour mappings at each time step, the overall direction of the cell track can vary over time.

The polarized cell tracks under varying regularization schemes are shown in S7 Fig. As mentioned above, clustering and thinning effects of virtual markers were prominent for $\lambda_{\text{reg}} = 0.01$. For the other choices of $\lambda_{\text{reg}}$, we observed that all kymographs are barely affected at all, resulting in similar contour dynamics (top right). Major differences in the cell's direction are noticed only at the beginning of the cell track when a stable distribution of virtual markers is not yet reached.

In S8 and S9 Figs, we present simulated non-polarized and polarized cell tracks for varying temporal resolution $\delta t = 0.25, 0.5, 1, 2, 2.5, 3.\bar{3}s$ and fixed regularization parameter $\lambda_{\text{reg}} = 10$. We observed that the local motion kymograph and, hence, the general contour dynamics are not substantially affected. In some cases, the overall direction of the contour dynamics was altered due to small differences at the beginning of the track.

In summary, we observed that by varying the regularization parameter within the range $\lambda_{\text{reg}} \in [0.1, 10]$, the overall contour dynamics are barely affected. The same observation was made for a varying temporal resolution. Based on the above studies, we have chosen a medium regularization $\lambda_{\text{reg}} = 10$ and a relatively dense temporal resolution of $\delta t = 0.5s$ for the following simulations.

**Long-time simulations are stable and show normal diffusive behavior.** To demonstrate the capability of our model to produce stable contour dynamics even for a longer time period, we simulated a variety of cell tracks under both scenarios, the non-polarized and the polarized case as described above. Furthermore, by showing that qualitatively and quantitatively different cell tracks can be generated with our model, we provide a rationale for the inference approach in the following section, where we applied the model to experimental cell tracks and different types of locomotion.

In Fig 6, we present the diffusive behavior of 50 cell tracks, respectively, for both polarization scenarios. For each scenario, a selection of ten cell tracks is highlighted in different colors (see panels (A) and (F)). The center of mass trajectories of these cell tracks for $T = 500s$ are displayed in panels (B) and (G). Furthermore, an enlarged excerpt of panel (B) is shown in panel (C). The root mean squared displacements (RMSD) of the trajectories are depicted as gray circles at time steps $t = 0, 100, \ldots, 500s$. The corresponding mean squared displacement (MSD) is shown in panels (D) and (H), indicating normal diffusion for the non-polarized case and a ballistic regime (so-called superdiffusion) for the polarized case for the first 500 seconds. In panels (E) and (I), the MSD is displayed for a longer time period of $T = 10000s$, clearly showing a linear relation between time and MSD indicating normal diffusive behavior also for the polarized case, see also S10 Fig, where the center of mass trajectories of the polarized case is presented for the entire time period of $T = 10000s$. Again, the RMSD is indicated as gray circles ($\delta t = 2000s$). Based on linear fits (red dashed lines), we computed the diffusion coefficients: $D \approx 0.16\mu m^2/s$ for the non-polarized case and $D \approx 34.33\mu m^2/s$ for the polarized case. In addition to the diffusion coefficient, a persistence time $P$ can be computed by fitting the mean squared displacements (MSD) to Fürth's formula (dimension n = 2):

$$\text{MSD}(t) = 2nD(t - P(1 - e^{-t/P})),$$

a standard reference model, where the trajectories are based on an Ornstein-Uhlenbeck process [57, 89]. By doing so, we determined a mean persistence time of 76.8min for the non-polarized case and 113.5min for the polarized case. The mean instantaneous speed of the center of mass

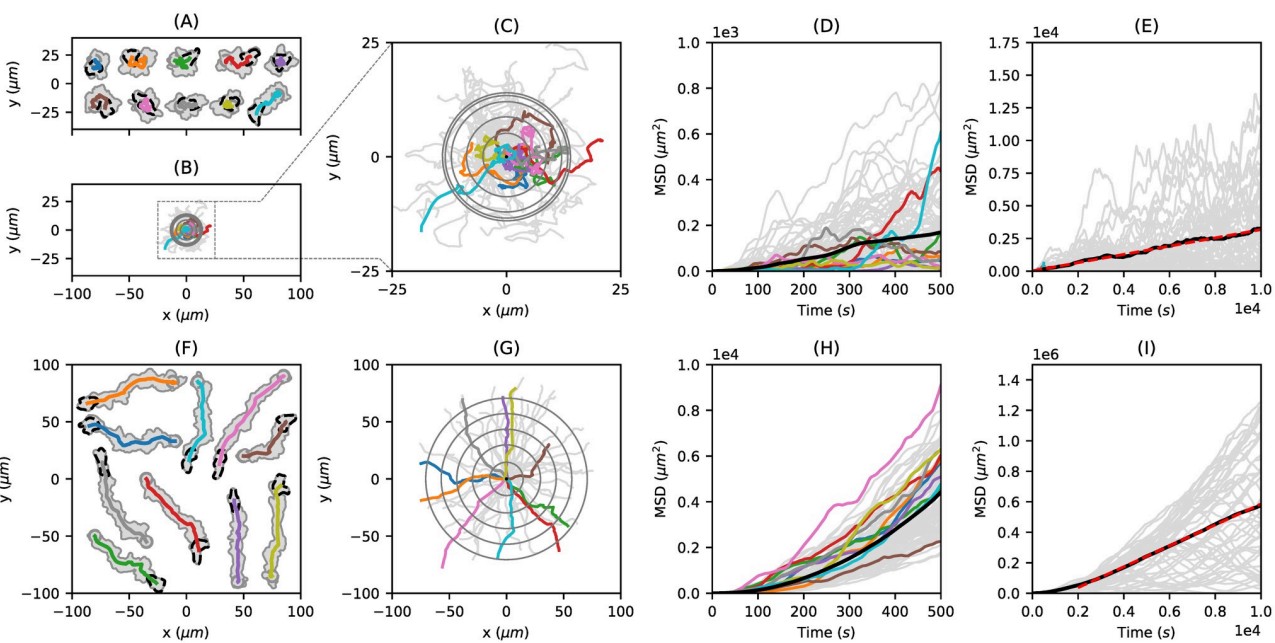

**Fig 6.** Diffusion analysis of artificial cell tracks generated from a non-polarized test scenario (top row) and a polarized test scenario (bottom row). (**A, F**) For each scenario, ten out of 50 exemplary cell tracks are shown. (**B, C, G**) Center of mass trajectories (colored lines) with root mean squared displacement (grey circles) at different time points with $\Delta t = 100s$. While panel (B) and (G) are scaled equally, panel (C) is an enlarged excerpt of panel (B).(**D, H**) Corresponding MSD of these trajectories (bold black line) for medium time spans of $T = 500s$.(**E, I**) MSD for longer time spans of $T = 10000s$ with linear fit (dashed red line).

is $3.9\mu m/\text{min}$ for the non-polarized case, whereas it is more than doubled for the polarized case ($8.7\mu m/\text{min}$). This observation is in accordance with previous studies where the existence of a universal coupling between cell persistence and cell speed has been shown [89].

As mentioned earlier, due to the stochastic nature of our model, noise-induced self-intersections cannot be ruled out explicitly, but get eliminated over time under the influence of the APCSF. However, in all our simulation studies, we did not observe self-intersections at any time.

In S4 and S5 Videos, the contour dynamics of the respective cell tracks from Fig 6A and 6F are shown.

## Inferring the protrusion component from experimental data

In the first part, we used our model to simulate a variety of different cell tracks based on stable and, in particular, realistic contour dynamics. In the second part, we will now apply our model to experimental data to analyze cell tracks on the individual level and to classify them based on different types of locomotion: the amoeboid type and a so-called fan-shaped type. The underlying microscopy imaging data was published in [90]. As mentioned earlier, the APCSF and AAF from Eq (16) only depend on the current contour and deterministic quantities such as contour curvature, arc length, and area. Hence, the remaining component $f_{\text{prot}}$ that propagates one contour to the consecutive one, can be determined explicitly. For a sequence of experimental cell contours, this approach enables us to infer the different model components for a given set of model weights; see S1 Text for more details on how to estimate these parameters.

In Fig 7A, contour dynamics of *D. discoideum* are displayed for a time period of $T = 500s$ and a frame rate of $\delta t = 1s$. This cell track is based on fluorescence microscopy data (mRFP tagged LifeAct, white to green color scheme), from which the cell contours are segmented

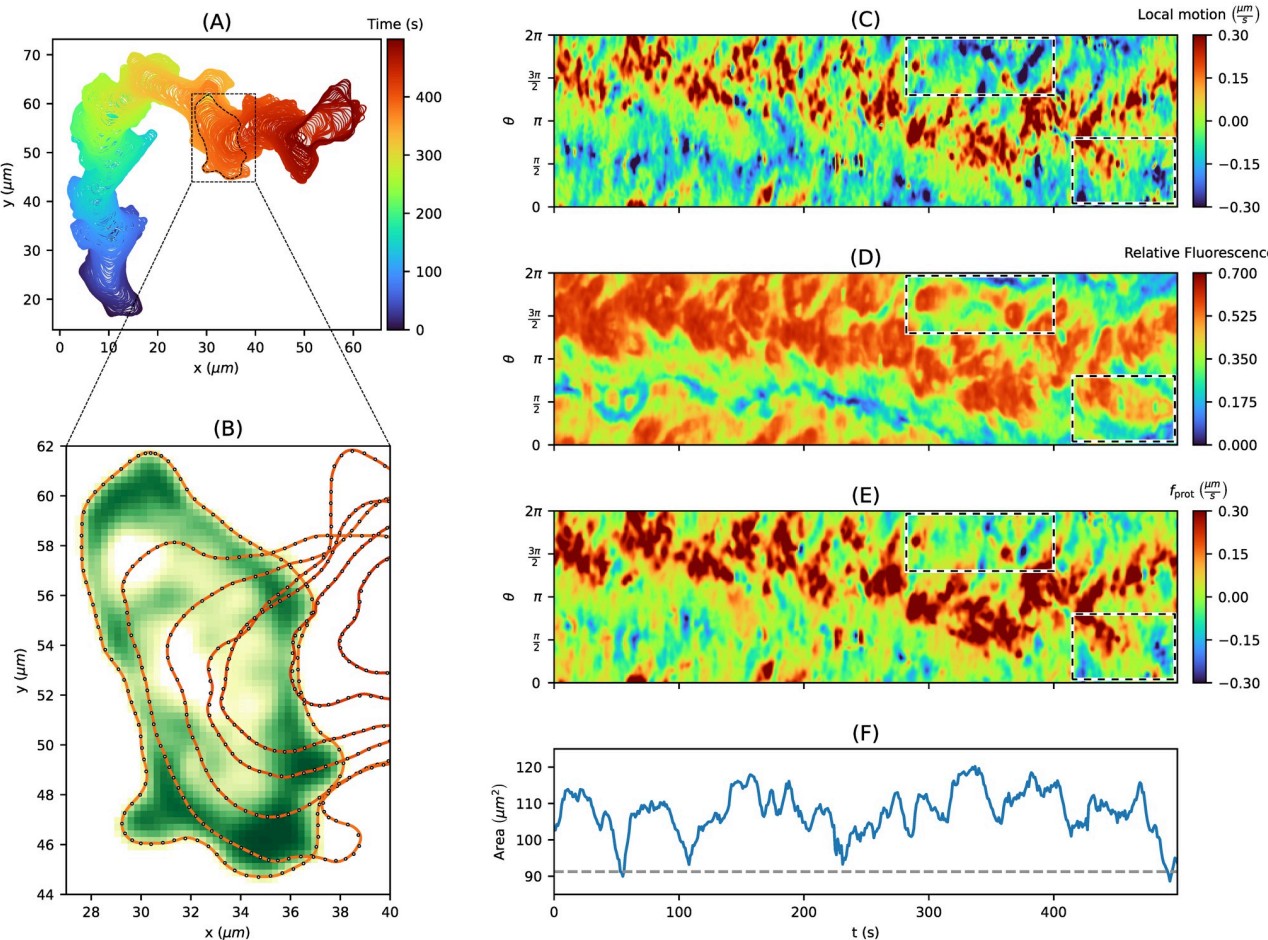

**Fig 7. Inferring protrusion component from *D. discoideum* cell track.** (A) Persistently motile cell track for $T = 500s$. (B) Microscopy image with fluorescence intensity (mRFP tagged LifeAct as marker for filamentous actin, white to green color scheme) and segmented cell contours (red lines, every tenth shown). (C) Local motion kymograph showing expansions (red areas) and retractions (blue areas). (D) Relative fluorescence intensity as in panel (B) with regions of high and low F-actin density displayed in red and blue, respectively. (E) The underlying protrusion component inferred from our model for a given set of model parameters. The resulting propagation of virtual markers from one contour to the next one is depicted as black circles in panel (B). (F) The contour area with predefined reference area $A_{\text{ref}} = 91.23\mu m^2$ (dashed gray line). Finally, regions of interest are displayed as black and white dashed boxes.

(colored lines), see panel (B). As mentioned, we have chosen the 1st percentile of the entire area time series as underlying reference area, i.e., $A_{\text{ref}} = 91.23\mu m^2$. By following the approach from S1 Text, we estimated the following model weights: $w_{\text{prot}} = 6.634\mu m^2/s, w_{\text{APCSF}} = 0.057\mu m^2/s$, and $w_{\text{AAF}} = 3.532\mu m/s$. In comparison to parameter values estimated for other cell tracks, we observed that $w_{\text{prot}}$ and $w_{\text{AAF}}$ are relatively large while $w_{\text{APCSF}}$ took a medium value. This observation coincides with the following cell track characteristics: a fast and persistent movement, an area time series within a smaller range $90\mu m^2 < A < 120\mu m^2$ (see panel (F)), and contour dynamics showing an intermediately regularized curvature. The accuracy of our computational approach is demonstrated in panel (B), with virtual markers (black circles) being effectively propagated onto consecutive contours. In the following, we compare the inferred protrusion component (panel (E)) to common biomarkers: the local motion of the membrane (panel (C)) and the F-actin density/fluorescence intensity close to the membrane (panel (D)) based on the fluorescence microscopy data as shown in panel (B).

**Inferred protrusion component in comparison to other biomarkers.** In contrast to the commonly used local motion which reflects the entire contour dynamics, i.e, protrusions, retractions, and minor membrane fluctuations, the proposed model separates the creation of protrusions, which depend on $f_{\mathrm{prot}}$, and retractions, which depend on $f_{\mathrm{APCSF}}$ and $f_{\mathrm{AAF}}$. We expect the protrusion component to be correlated to the F-actin density near the membrane. However, here we have chosen images, where the fluorescence signal saturates at high F-actin levels to facilitate segmentation of the cell contours. Thus, by following this approach, the resulting fluorescence intensity of the F-actin density provides less details.

In Fig 7, kymographs of all three quantities are displayed: the local motion (panel (C)), the fluorescence intensity (panel (D)), and the inferred protrusion component (panel (E)). We expect that a kymograph of a successfully inferred protrusion component would be strictly positive and only indicate regions where protrusions occurred. The remaining retractions are then covered by the other two components of our model: the AAF and the APCSF. In the local motion kymograph, local protrusions (red regions) and retractions (blue regions) are nicely shown. Due to a saturating fluorescence signal at high F-actin levels, we expect fewer details in the resulting fluorescence intensity kymograph for higher intensity levels. Indeed, the corresponding kymograph in panel (D) seems to be smoother than the local motion kymograph in panel (C), especially, for higher relative fluorescence intensity levels at around 0.6. The protrusion component kymograph in panel (E) mainly displays protrusive areas (red regions) with only very few negative regions (blue).

In S11 Fig, we present our approach to measuring the fluorescence intensity near the membrane by averaging over ellipses along the cell contour. To improve the quality of the fluorescence intensity kymograph, we post-processed the recorded microscopy images by using a (tenfold) image upsampling, i.e., we smoothed all microscopy images by increasing the number of pixels tenfold. However, the resulting kymograph showed no differences from the kymograph based on the original image data.

In S12 Fig, the protrusion component from Fig 7 is displayed as well as the remaining components $f_{\mathrm{APCSF}}$ and $f_{\mathrm{AAF}}$. As the underlying set of model weights, we used the same estimates mentioned above. The contributions of all three model components to the overall dynamics are shown for each time step/contour. In this context, we observed that the APCSF accounts for approximately 10% of the overall dynamics. In contrast, the AAF and the protrusion component are more prominent accounting for approximately 40% and 50%, respectively. This may indicate that the APCSF mainly resolves a smaller number of strongly curved contour segments, whereas slower contour retractions at the rear of the cell are controlled by the AAF. As mentioned earlier, the protrusion component kymograph mainly consists of positive areas (red regions). Since the local motion comprises the summarized information of all three model components and due to the different portions of the AAF and APCSF to the overall dynamics (40% vs. 10%), we conclude that the retractions (blue regions) in the local motion kymograph in Fig 7 are primarily captured by the AAF and to a lesser extent by the APCSF.

In addition, we present kymographs of the same quantities for an alternative set of model parameters, where positive values of $f_{\mathrm{prot}}$ are more favored; see S1 Text for more details. For this case, $f_{\mathrm{prot}}$ and $f_{\mathrm{AAF}}$ accounted for the overall dynamics equally, whereas $f_{\mathrm{APCSF}}$ was observed to be negligible most of the time.

**Impact of *a priori* parameter choices on the analysis of experimental data.** The F-actin density near the membrane is often interpreted as a marker of protrusive activity. Thus, we would expect some correlation between the F-actin density and the protrusion component in our model. Since the latter is influenced by the choice of the model weights of the two other model components AAF and APCSF, the question arises to what extent the correlation depends on the *a priori* choices of $w_{\mathrm{AAF}}$ and $w_{\mathrm{APCSF}}$.

In S1 Text, we rewrite the model equations from Eq (16) with respect to relative model weights instead of the absolute weights $w_{\text{prot}}$, $w_{\text{APCSF}}$, and $w_{\text{AAF}}$. An impact analysis of the varying model weights on the inferred protrusion component is also included in S1 Text. The inferred protrusion component kymographs resulting from this impact analysis are shown in S13 Fig. For the different configurations of model weights, we computed the Pearson correlation coefficient between the protrusion component and the Fluorescence intensity of the F-actin density, see S13 Fig.

Briefly, we observed that the protrusion component inferred by the model is correlated to the underlying F-actin density. While the protrusion component is affected significantly by the parameter choice, the impact on the Pearson correlation coefficient is minor. In the case of the estimated model weights used in Fig 7, a Pearson correlation coefficient of $\rho = 0.47$ was computed, which is close to the Pearson correlation for the best fit ($\rho = 0.49$), see S1 Text and S13 Fig for details.

**Classification of contour dynamics based on cell motility types.** In the first part of this work, we have shown that our model can be used to simulate a variety of different contour dynamics. Here, we demonstrate that our model can also be applied to different experimental cell tracks. By inferring the protrusion component and estimating the underlying model weights, we analyzed contour dynamics on the individual level and classified them based on two different types of locomotion: the amoeboid type and the so-called fan-shaped type. Again, the contour dynamics were derived from fluorescence images of *D. discoideum*, where frames are recorded with a temporal resolution of $\delta t = 1s$ (amoeboid type) and $\delta t = 4s$ (fan-shaped type).

The contour parameters $r_{\text{cont}} = 0.6$, $\sigma_{\text{noise}} = 0.05$ were chosen as in the simulation part, for more information see S1 Text of [84]. The reference area $A_{\text{ref}}$ was chosen individually for each cell track based on the 1st percentile of the entire time series of the contour area (gray dashed line): 62.43, 122.99, and $64.61\mu m^2$ for the amoeboid cell tracks; 142.15, 203.45, and $189.24\mu m^2$ for the fan-shaped cells. Moreover, the model weights were estimated for each cell track individually, described as in S1 Text.

In Fig 8, three tracks with an amoeboid motion as in Fig 7 are presented. In contrast to the local motion (first kymograph row), the protrusion component (third kymograph row) contains only a few negative (blue) regions, which means that the APCSF and AAF capture most of the contour retractions successfully. Furthermore, we see strong correlations between all three quantities. However, the protrusive regions in the fluorescence intensity kymographs are larger than for the other two kymographs, which is a direct consequence of detector saturation during fluorescence imaging. From the estimated model weights, we can infer that the third cell track is more motile ($w_{\text{prot}} = 6.505\mu m^2/s$) than the other two cell tracks ($w_{\text{prot}} = 4.513\mu m^2/s$) for both), which coincides with the contour dynamics displayed on top. Furthermore, we see that the AAF weight $w_{\text{AAF}}$ directly influences the variability of the area time series, i.e., less variability for the second cell track ($w_{\text{AAF}} = 1.429\mu m/s$) vs. more variability for the first and third cell track ($w_{\text{AAF}} = 0.902\mu m/s$ and $0.676\mu m/s$). Finally, we can observe that the elongated contour shape of the second and third cell track coincide with higher APCSF weights ($w_{\text{APCSF}} = 0.048\mu m^2/s$ and $0.070\mu m^2/s$) compared to the smaller weight of the first cell track ($w_{\text{APCSF}} = 0.023\mu m^2/s$).

In the case of the fan-shaped cells in Fig 9, we see clear blue stripes in the local motion kymograph at the rear of the cell, which corresponds to a smaller F-actin density in the fluorescence intensity kymograph. Because of the characteristic kidney-shaped cell contour, our model predicts a negative protrusion component at the cell's rear also displayed as a light blue stripe in the kymographs. This indicates that the APCSF and the AAF were not able to fully capture this retraction. Since the APCSF evolves every contour to a circle, it counteracts the

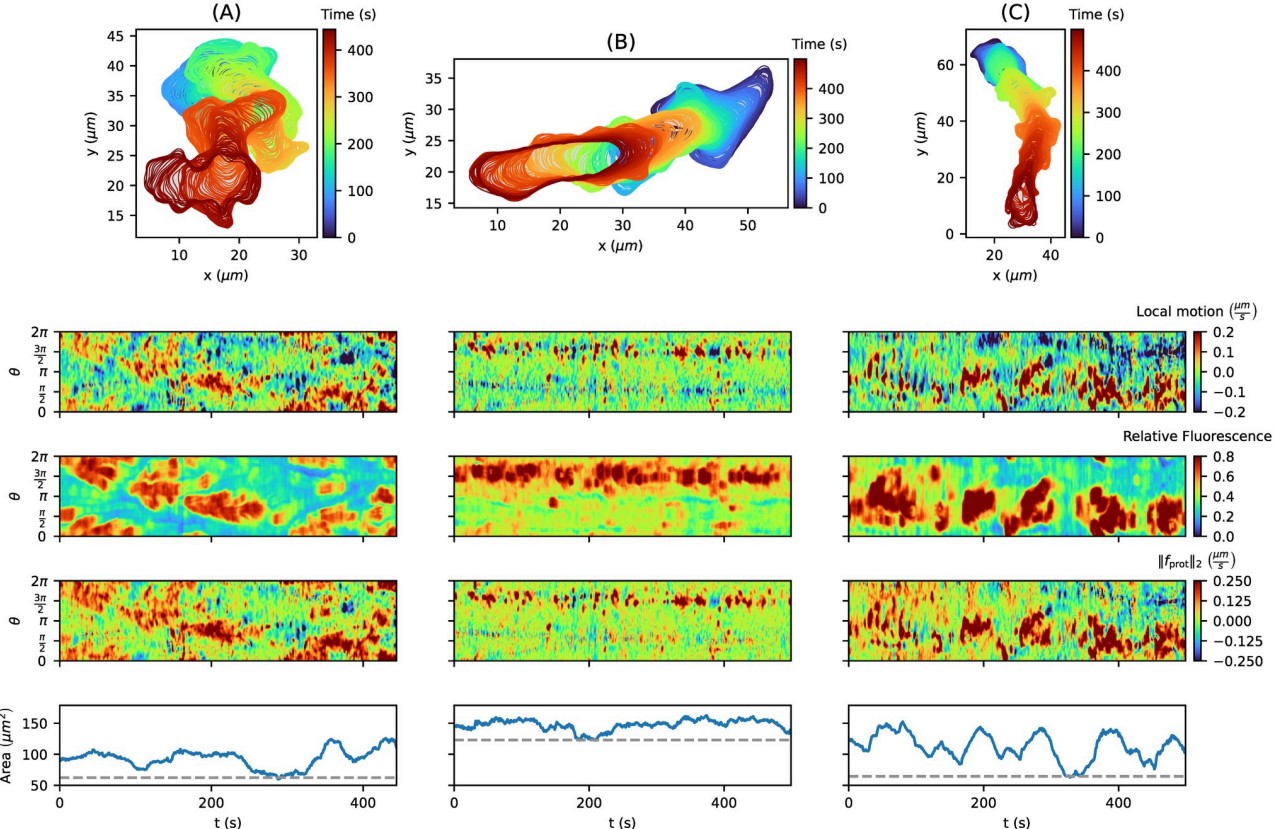

**Fig 8. Collection of three *D. discoideum* tracks driven by the standard amoeboid type of cell motion with corresponding kymographs: Local motion, relative fluorescence intensity, the underlying protrusion component inferred from our model, and the contour area.** For each cell track, the 1st percentile of the individual area time series (last row) was chosen as the reference area (gray dashed line) in our model: 62.43, 122.99, and 64.61 $\mu m^2$, respectively. The model weights ($w_{\text{prot}}$, $w_{\text{APCSF}}$, $w_{\text{AAF}}$) were estimated individually for each cell track: (4.513, 0.023, 0.902), (4.513, 0.048, 1.429), and (6.505, 0.070, 0.676), respectively from **(A)** to **(C)**.

characteristic concave-shaped contour of the cell. For this reason, we expect the estimates of $w_{\text{APCSF}}$ to be lower than for amoeboid locomotion. Indeed, the APCSF weight was estimated to be zero for all three cells. Therefore, by inferring the impact of the APCSF, we have shown that our model is capable of distinguishing fan-shaped cells from standard amoeboid cells. For the second and third cell track, similar weights $w_{\text{prot}} = 4.485 \mu m^2/s$ vs. $4.428 \mu m^2/s$ and $w_{\text{AAF}} = 1.844 \mu m/s$ vs. $1.76 \mu m/s$ were estimated, which is in accordance with the similar contour dynamics reflected by comparable mean centroid velocities of $0.091 \mu m/s$ and $0.099 \mu m/s$, respectively. In contrast, we estimated a lower protrusion component weight $w_{\text{prot}} = 3.121 \mu m^2/s$ for the first cell track, which is in agreement with a smaller mean centroid velocity of $0.069 \mu m/s$. Moreover, we estimated a smaller AAF weight $w_{\text{AAF}} = 0.911 \mu m/s$, which might be a result of the ever-increasing contour area of this track ($130 \mu m^2$ to $217 \mu m^2$).

## Discussion

We developed a novel model to simulate and analyze the contour dynamics of amoeboid cell migration. The contour dynamics model is based on three components: (1) a stochastic protrusion component based on a self-exciting Poisson point process also known as Hawkes process, (2) an area-preserving curve-shortening flow (APCSF) to regularize the contour arc

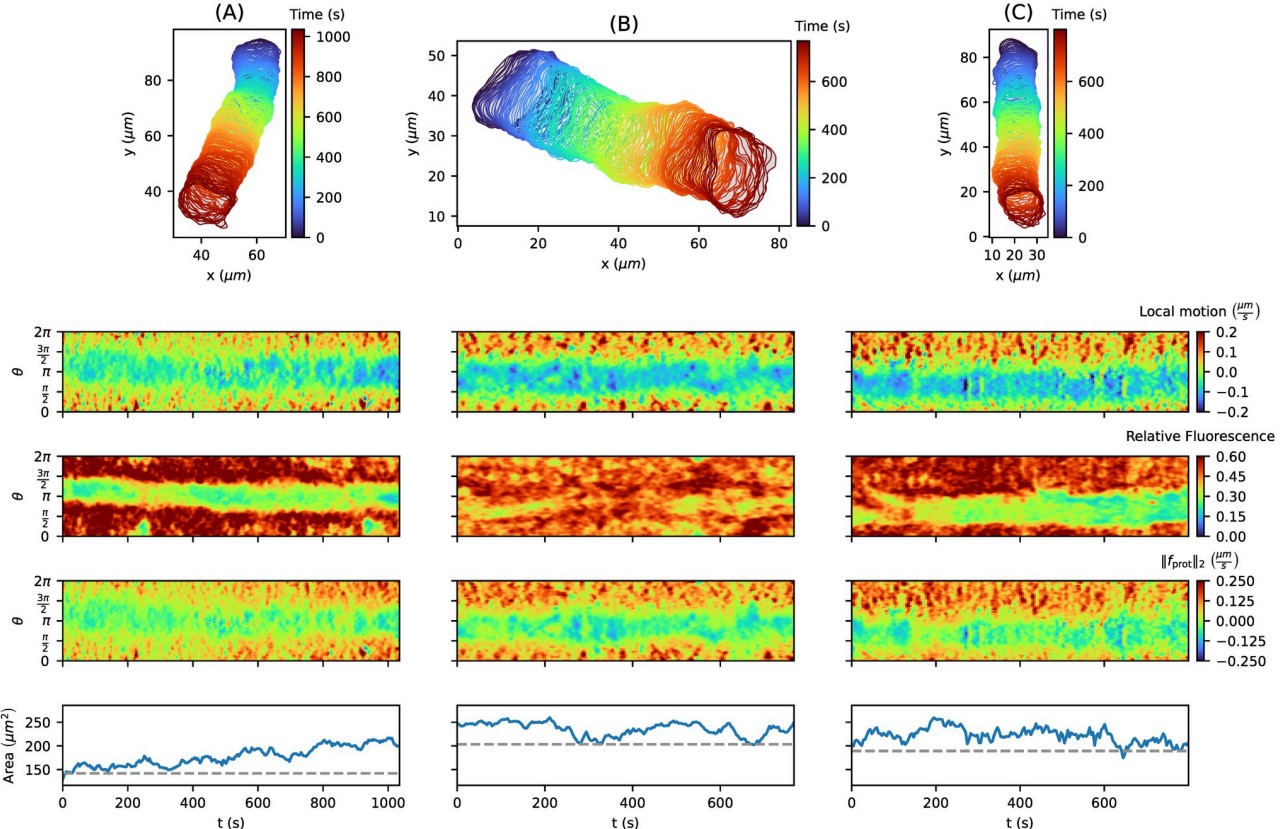

**Fig 9. Collection of three *D. discoideum* tracks driven by a fan-shaped type of cell motion with corresponding kymographs: Local motion, relative fluorescence intensity, the underlying protrusion component inferred from our model, and the contour area.** For each cell track, the 1st percentile of the individual area time series (last row) was chosen as the reference area (gray dashed line) in our model: 142.15, 203.45, and 189.24 $\mu m^2$, respectively. The model weights ($w_{\mathrm{prot}}$, $w_{\mathrm{APCSF}}$, $w_{\mathrm{AAF}}$) were estimated individually for each cell track: (3.121, 0.000, 0.911), (4.485, 0.000, 1.844), and (4.428, 0.000, 1.760), respectively from **(A)** to **(C)**.

length, (3) and a further geometric flow introduced as Area Adjustment Flow (AAF) to control the contour area. While the first component controls the forward movement of the cell, the latter two components control contour retractions, occurring most often at the rear of the cell.

First, we have shown that our model is capable of generating a variety of cell tracks with different spatio-temporal patterns. We simulated non-polarized as well as polarized contour dynamics that look realistic and remain stable even for longer time periods. Secondly, we applied our model to experimental cell tracks to analyze key motility characteristics based on the inference of the three model components and the estimation of the underlying model component weights. By examining the correlation between the inferred protrusion component and the fluorescence intensity reflecting the F-actin density, we demonstrated that the creation of pseudopods is correctly accounted for by the protrusion component of our model. Furthermore, our model was capable of decoupling the explorative protrusions from the slower contour retractions, which in our model are solely based on the APCSF and the AAF. By estimating the model weights of each component, we demonstrated a simple approach to classify cells based on two locomotion types: the amoeboid and a so-called fan-shaped type.

The simplest approaches to model amoeboid cell motility focus on the motion of the center of mass of the cell [60–62]. Often, the center of mass trajectory is modeled by a Fokker-Planck equation [27] or a Langevin equation [57–61]. Due to a lower model complexity, even a large

number of cell tracks can be generated relatively easily and fast. These models are then used to examine statistical quantities such as the diffusion coefficient, the persistence time, and the drift velocity. While our model focuses on contour dynamics, the center of mass trajectory and the corresponding statistics can be derived easily. For example, we determined a diffusion rate of $D = 0.16 \mu m^2/s$ in our test case of simulated non-polarized cell tracks which is in accordance with experimental measurements [27, 91].

Other approaches include biochemical models focusing on intracellular signaling molecules and cytoskeletal components [24, 25] or specific biophysical mechanisms such as cell polarization during chemotaxis [18, 19]. Some models combine intracellular processes with specific membrane patterns such as the formation of pseudopods [22] or changes of the cell shape in general [92]. Furthermore, phase-field models are often used in which the transition of different states such as solid/liquid or interior/exterior of the cell are described [26, 35, 36]. In these models, biochemical markers such as the F-actin density are often propagated in time and space and drive displacements of the cell membrane [13, 34].

In contrast to most of the existing approaches which focus on the microscopic level of chemical and biophysical processes during cell migration, our model describes amoeboid motility on a macroscopic level. By relying on the deformation of an elastic object, namely the cell contour, it has a mechanistic basis. Usually, mechanistic approaches are based on the interplay of multiple forces affecting the cell from the outside but also from within [42, 43, 48]. However, some mechanistic models rely on multiple entangled subprocesses making it more difficult to achieve a direct causality between a chosen parameter regime and a specific motility behavior [48]. Here, our model differs substantially due to its dependency on three components only and a low number of parameters: 4 model parameters and 4–5 additional parameters regarding the stochastic process. Desired motility characteristics such as the number, duration, and size of protrusions, the velocity of the evolving contour, or the cell polarization can be easily achieved based on an appropriate specification of the underlying parameter regime. In the long term, our aim is to add a biochemical component to our model by using the F-actin density as an input of the protrusion component. This way, our model can be potentially used to infer biological insights on an intracellular level, which is, however, beyond the scope of this paper. As a first step in this direction, we compared the inferred protrusion component of experimental cell tracks with the underlying F-actin density, laying the ground for such a data-driven approach.

Motivated by the biological insight that the location of protrusions depends on previous protrusions [17] and is triggered by a cascade of physiological events affecting the growth of the actin filament network [9], we have chosen a Hawkes process as the underlying stochastic process. Due to its self-exciting property, the Hawkes process is capable of generating multiple explorative protrusions in temporal and spatial proximity with subsequent reorientation phases characterized by a temporary decline of cell motility. Compared to other processes such as the Ornstein-Uhlenbeck process or a standard Poisson process, the Hawkes process is therefore advantageous to model membrane protrusions.

In a previous approach, we used phase-field equations to model amoeboid cell motility [13, 34]. This phase-field model also consists of three terms: surface tension, volume conservation, and active tension which correspond to the three components of this work, i.e., the APCSF, the AAF, and the protrusion component, respectively. The phase-field model describes the changes of a two-dimensional biochemical component, reacting and diffusing inside the cell, and leading to a propagation of the entire cell. In contrast, the present work focuses on the propagation of a one-dimensional object only, namely the cell contour, further reducing the complexity of the model. Furthermore, the present model does not rely on the underlying concentration of biomarkers, e.g., the F-actin density, but can be used to infer information about

the protrusions of experimental cell tracks. Due to the area-regularizing effect of the AAF, the cell area is preserved within a specific range, which is similarly done by the volume conservation terms in different phase-field models [34, 35, 93]. In contrast to our previous phase-field model, where new protrusions were generated by an Ornstein-Uhlenbeck process driven by Gaussian white noise, the focus of this work is mainly on the Hawkes process due to its self-exciting nature. Finally, our phase-field model was used to study more complex processes such as cell division or the interaction and movement of multiple cells [94, 95].

By focusing on the contour dynamics to model cell motility, further assumptions regarding the mapping of virtual markers along time and space were necessary. We assumed that the transport of the underlying protrusion process is based on the concept of regularizing contour flows which were previously introduced in [84]. Other approaches to determining contour/virtual marker mappings include electrostatic field equations [85], level-set methods [85, 86], or mechanistic spring equations [87]. All of the above approaches, including ours, share the same problem that the mapping of consecutive contours is not defined *a priori*. We have shown that our contour mapping assumption has a minor effect on the overall contour dynamics and we provided a reasonable range of the underlying regularization parameter $\lambda_{\mathrm{reg}}$ for which simulated contour dynamics are stable.

So far, only few approaches offer a full integration of the numerical model and experimental measurements, e.g., for fibroblast migration [51]. Our model provides a full and automatized integration of experimental measurements due to its capability of inferring protrusion and retraction components individually and by offering a simple and straightforward estimation of the underlying model weights. The most common approach to tune computational models with respect to experimental data is to perform a sensitivity analysis [48, 96–98]. In this context, different parameter regimes are chosen to simulate different motility behaviors and to determine the importance of each parameter on the simulated outcome. With a similar approach, we have shown that the inference of the protrusion component by our model is stable for varying model weights $w_{\mathrm{prot}}$ and $w_{\mathrm{AAF}}$. However, if the APCSF is too prominent, the inferred protrusion component is negatively affected, indicated by a weaker correlation with the underlying F-actin density. Regarding the classification of different motility types, we examined contour dynamics based on a single locomotion type (either amoeboid or fan-shaped). However, recent studies report spontaneous switching between these two locomotion types [82].

In summary, the proposed model sets new standards in simulating stable and, in particular, realistic contour dynamics. Due to a fast and straightforward parameter estimation and an automatized approach to infer protrusion and retraction characteristics, the model can be used to analyze experimental cell tracks on the individual level, to classify them, and as a general comparison tool for experimental contour data as well as contour dynamics generated by different cell motility models.

## Supporting information

**S1 Text. Supporting formulas and computations.**
(PDF)

**S1 Fig. Artificial contour dynamics based on our model with a comparison of the underlying contour propagation and contour mapping.**
(PDF)

**S2 Fig. Comparison of Poisson kernel function and its normalized version with varying Poisson radius parameter *r*.**
(PDF)

**S3 Fig. Illustration of kernel functions used to generate artificial cell tracks driven by a Hawkes process.**
(PDF)

**S4 Fig. Collection of five non-polarized cell tracks based on Poisson point processes as protrusion process and the corresponding kymographs displayed as in Fig 3.**
(PDF)

**S5 Fig. Collection of five polarized cell tracks based on Poisson point processes as protrusion process and the corresponding kymographs displayed as in Fig 4.**
(PDF)

**S6 Fig. Parameter study with varying regularization parameter $\lambda_{\mathbf{reg}}$ during non-polarized cell motility.**
(PDF)

**S7 Fig. Parameter study with varying regularization parameter $\lambda_{\mathbf{reg}}$ during polarized cell motility.**
(PDF)

**S8 Fig. Comparison of local motion kymographs for different non-polarized cell tracks generated with different temporal resolutions: $\delta t = 0.25, 0.5, 1, 2, 2.5, 3.\bar{3}s$.**
(PDF)

**S9 Fig. Comparison of local motion kymographs for different polarized cell tracks generated with different temporal resolutions: $\delta t = 0.25, 0.5, 1, 2, 2.5, 3.\bar{3}s$.**
(PDF)

**S10 Fig. Center of mass trajectories of polarized cell tracks over a longer time period ($T = 10000s$) with corresponding MSD as in Fig 6.**
(PDF)

**S11 Fig. Computation of relative fluorescence intensity for experimental microscopy data and tenfold upsampled data via ellipses along the cell contour.**
(PDF)

**S12 Fig. Model components extracted from experimental cell track of Fig 7 and their proportion on the overall velocity of the contour dynamics for two pairs of model weights.** The model weights were estimated by minimizing sums of squared residuals: *S* (left column) and $S^+$ (right column), see S1 Text for more details.
(PDF)

**S13 Fig. Protrusion component $f_{\mathbf{prot}}$ extracted from the experimental cell track of Fig 7 for varying relative weights $r_{\mathbf{prot}} \in \{0.05, 0.5, 0.8\}$ (vertical axis) and $r_{\mathbf{APCSF}} \in \{0.01, 0.05, 0.1\}$ (horizontal axis) as well as varying overall velocity parameter $w_f \in \{1, 5, 10, 20\}$ (page axis).**
(PDF)

**S1 Video. Contour dynamics with corresponding kymographs of a non-polarized cell track based on a Hawkes process as shown in Fig 3.** The point events generated by the Hawkes process are depicted as white circles. The cell track is displayed at a fivefold speed.
(MP4)

**S2 Video. Contour dynamics with corresponding kymographs of a polarized cell track based on a Hawkes process as shown in Fig 4.** The point events generated by the Hawkes

process are depicted as white circles. The cell track is displayed at a fivefold speed.
(MP4)

**S3 Video. Contour dynamics for different modifications of the underlying Ornstein-Uhlenbeck process as protrusion process.** The cell tracks are displayed at tenfold speed.
(MP4)

**S4 Video. Contour dynamics of non-polarized cell tracks based on a Hawkes process as shown in Fig 6.** The cell tracks are displayed at tenfold speed.
(MP4)

**S5 Video. Contour dynamics of polarized cell tracks based on a Hawkes process as shown in Fig 6.** The cell tracks are displayed at tenfold speed.
(MP4)

## Acknowledgments

We want to thank Elena Mäder-Baumdicker for the very constructive feedback regarding the use of the APCSF. Furthermore, we acknowledge Karl-Michael Schindler and Bertram Arnold for fruitful and stimulating discussions.

## Author Contributions

**Conceptualization:** Daniel Schindler, Carsten Beta, Wilhelm Huisinga, Matthias Holschneider.

**Formal analysis:** Daniel Schindler.

**Funding acquisition:** Carsten Beta, Wilhelm Huisinga, Matthias Holschneider.

**Investigation:** Daniel Schindler, Ted Moldenhawer.

**Methodology:** Daniel Schindler, Carsten Beta, Wilhelm Huisinga, Matthias Holschneider.

**Software:** Daniel Schindler, Ted Moldenhawer.

**Supervision:** Carsten Beta, Wilhelm Huisinga, Matthias Holschneider.

**Validation:** Daniel Schindler.

**Visualization:** Daniel Schindler.

**Writing – original draft:** Daniel Schindler.

**Writing – review & editing:** Daniel Schindler, Ted Moldenhawer, Carsten Beta, Wilhelm Huisinga, Matthias Holschneider.

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
