## [Decision Letter · Decision Letter 0]

29 Sep 2023

PONE-D-23-25151Three-component contour dynamics model to simulate and analyze amoeboid cell motility in two dimensionsPLOS ONE

Dear Dr. Holschneider,

Thank you for submitting your manuscript to PLOS ONE. After careful consideration, we feel that it has merit but does not fully meet PLOS ONE’s publication criteria as it currently stands. Therefore, we invite you to submit a revised version of the manuscript that addresses the points raised during the review process.

We look forward to receiving your revised manuscript.

Kind regards,

Xiakun Chu, Ph.D.

Academic Editor

PLOS ONE

Journal Requirements:

 "The research of DS and TM has been partially funded by the Deutsche Forschungsgemeinschaft (DFG)- Project-ID 318763901 - SFB1294."

 "The research of DS and TM has been partially funded by the Deutsche Forschungsgemeinschaft (DFG)- Project-ID 318763901 - SFB1294."

6. We are unable to open your Supporting Information file S2_Fig.eps and S10_Fig.eps. Please kindly revise as necessary and re-upload.

Reviewers' comments:

Reviewer's Responses to Questions

**Comments to the Author**

1. Is the manuscript technically sound, and do the data support the conclusions?

Reviewer #1: Yes

Reviewer #2: Yes

2. Has the statistical analysis been performed appropriately and rigorously? 

Reviewer #1: Yes

Reviewer #2: Yes

3. Have the authors made all data underlying the findings in their manuscript fully available?

Reviewer #1: Yes

Reviewer #2: Yes

4. Is the manuscript presented in an intelligible fashion and written in standard English?

Reviewer #1: Yes

Reviewer #2: No

5. Review Comments to the Author

Reviewer #1: The authors have presented a model with which they can simulate cell motility with a clear relationship between the parameters and the resulting motility. The model is capable of displaying different motilities, resembling amoeboid and fan shaped motility.Their model can also be used to fit on experimental cell tracks and be used to estimate the underlying parameters for such tracks. From such fits, the model component for protrusion correlates with the fluorescence of actin, which is a nice result, as the actin fluorescence seems indicative of protrusions and are not considered in the fitting process.

Major points:

1 The authors have not mentioned the Cellular Potts model as a model for cell motility in their introduction. I think the Act model extension (https://doi.org/10.1371/journal.pcbi.1004280) deserves mentioning. Furthermore, I think the methods presented might be interesting to use as a comparison tool between other models, such as the CPM, and/or experimental data.

2 In equation 11, cell polarization is implemented. As far as I've understood this, the polarization angle is fixed. In what ways could Eq. 11 be adapted to allow for things like repolarization, for instance upon collision with other cells or obstacles?

3 Lines 543-552: I think it is also interesting to report the persistence time alongside the diffusion coefficient, which can easily be found by fitting Fürth's equation. Regarding the direct relation between parameters and motility, I wonder if this model is able to display the universal coupling between speed and persistence (https://doi.org/10.1016/j.cell.2015.01.056), or whether this is fully decoupled in this model? Maybe the authors would like to discuss on this?

Minor points:

4 Fig 2: some parameter values are given, are the remaining as stated in table 1?

5 Figure 5 For me it was not immediately clear that the colors in panel B and correspond to those in D and E. Also, I think that in the sentence "The same protrusion component is underlying in (B) and (B), respectively." one of the (B) is probably a (C).

6 Fig 6: It may be good to note that n=50 in the caption of this figure.

7 Figure 7: Fluoresence intensity of what fluorescent signal is being depicted in panel B?

8 Line 633: I found the notation of relative weight parameters confusing at first, as they look similar to the parameter r_cont and r_pol. Consider less similar parameter symbols.

9 Lines 648-671 and Fig S15, please specify which correlation coefficent was used.

10 The order in which the supplementary figures appear in the main text is not the same as their numbering.

11 Fig S1: some typos in caption

Reviewer #2: This article presents the authors’ remarkable modeling efforts toward accounting for amoeba cell motility. It uses a 2D framework focusing on contour dynamics and consisting of stochastic simulations and other deterministic driving mechanisms. It also offers a method to fit on experimental data. This work provides a unique view of this classical problem. The authors also decided to present alternative driving stochastic processes (simple Poisson process, O-U process). This is especially helpful in the article’s completeness in contrast to if they chose to present only the final successful story to make it superficially magical. However, I have non-negligible concerns about its readability, as well as some minor issues on the logics of wording. At PLOS One, objective assessment of logic and presentation is valued primarily, rather than subjective impact or popularity, so I hope the authors and editor(s) can take into account the following call-outs. Even though the piece of research is deemed rigorous and sound, It is my humble opinion that a major revision to this article fixing at least the majority of these concerns before the article can be published.

Details are attached in a separate file.

6. PLOS authors have the option to publish the peer review history of their article (what does this mean?). If published, this will include your full peer review and any attached files.

Reviewer #1: No

Reviewer #2: No

---

## [Author Response · Author response to Decision Letter 0]

1 Dec 2023

We highly appreciated the constructive comments and remarks by the reviewers and the editor. We have addressed all comments and remarks and revised the manuscript accordingly. A response letter with point-to-point answers to the reviewers’ comments is provided ("Response to Reviewers (revised draft): response_to_reviewers_plos_one.pdf").

---

## [Decision Letter · Decision Letter 1]

8 Jan 2024

Three-component contour dynamics model to simulate and analyze amoeboid cell motility in two dimensions

PONE-D-23-25151R1

Dear Dr. Holschneider,

We’re pleased to inform you that your manuscript has been judged scientifically suitable for publication and will be formally accepted for publication once it meets all outstanding technical requirements.

Kind regards,

Xiakun Chu, Ph.D.

Academic Editor

PLOS ONE

Additional Editor Comments (optional):

Reviewers' comments:

Reviewer's Responses to Questions

**Comments to the Author**

1. If the authors have adequately addressed your comments raised in a previous round of review and you feel that this manuscript is now acceptable for publication, you may indicate that here to bypass the “Comments to the Author” section, enter your conflict of interest statement in the “Confidential to Editor” section, and submit your "Accept" recommendation.

Reviewer #1: All comments have been addressed

Reviewer #2: All comments have been addressed

2. Is the manuscript technically sound, and do the data support the conclusions?

Reviewer #1: Yes

Reviewer #2: Yes

3. Has the statistical analysis been performed appropriately and rigorously? 

Reviewer #1: Yes

Reviewer #2: Yes

4. Have the authors made all data underlying the findings in their manuscript fully available?

Reviewer #1: Yes

Reviewer #2: Yes

5. Is the manuscript presented in an intelligible fashion and written in standard English?

Reviewer #1: Yes

Reviewer #2: Yes

6. Review Comments to the Author

Reviewer #1: (No Response)

Reviewer #2: I would like to thank the authors for their detailed response and revision to reviewers' comments. I also thank the authors for their tolerance of my nitpicking. All my concerns are adequately addressed and I hope the article will receive great readership.

7. PLOS authors have the option to publish the peer review history of their article (what does this mean?). If published, this will include your full peer review and any attached files.

Reviewer #1: No

Reviewer #2: No

---

## [Editor Report · Acceptance letter]

18 Jan 2024

PONE-D-23-25151R1 

PLOS ONE

Dear Dr. Holschneider, 

I'm pleased to inform you that your manuscript has been deemed suitable for publication in PLOS ONE. Congratulations! Your manuscript is now being handed over to our production team.

Kind regards, 

on behalf of

Dr. Xiakun Chu 

Academic Editor

PLOS ONE